# Association of close-range contact patterns with SARS-CoV-2: a household transmission study

Jackie Kleynhans[1,2]*, Lorenzo Dall'Amico[3], Laetitia Gauvin[3,4], Michele Tizzoni[3,5], Lucia Maloma[6], Sibongile Walaza[1,2], Neil A Martinson[6,7], Anne von Gottberg[1,8], Nicole Wolter[1,8], Mvuyo Makhasi[1,2], Cheryl Cohen[1,2], Ciro Cattuto[3,9], Stefano Tempia[1,2], SA-S-HTS Group

[1]Centre for Respiratory Diseases and Meningitis, National Institute for Communicable Diseases of the National Health Laboratory Service, Johannesburg, South Africa; [2]School of Public Health, Faculty of Health Sciences, University of the Witwatersrand, Johannesburg, South Africa; [3]ISI Foundation, Turin, Italy; [4]Institute for Research on Sustainable Development, Aubervilliers, France; [5]Department of Sociology and Social Research, University of Trento, Trento, Italy; [6]Perinatal HIV Research Unit, University of the Witwatersrand, Johannesburg, South Africa; [7]Johns Hopkins University Center for TB Research, Baltimore, United States; [8]School of Pathology, Faculty of Health Sciences, University of the Witwatersrand, Johannesburg, South Africa; [9]Department of Informatics, University of Turin, Turin, Italy

**\*For correspondence:**
jackiel@nicd.ac.za

**Group author details:**
SA-S-HTS Group See page 22

## Abstract

**Background:** Households are an important location for severe acute respiratory syndrome coronavirus 2 (SARS-CoV-2) transmission, especially during periods when travel and work was restricted to essential services. We aimed to assess the association of close-range contact patterns with SARS-CoV-2 transmission.

**Methods:** We deployed proximity sensors for two weeks to measure face-to-face interactions between household members after SARS-CoV-2 was identified in the household, in South Africa, 2020–2021. We calculated the duration, frequency, and average duration of close-range proximity events with SARS-CoV-2 index cases. We assessed the association of contact parameters with SARS-CoV-2 transmission using mixed effects logistic regression accounting for index and household member characteristics.

**Results:** We included 340 individuals (88 SARS-CoV-2 index cases and 252 household members). On multivariable analysis, factors associated with SARS-CoV-2 acquisition were index cases with minimum $C_t$ value <30 (aOR 16.8 95% CI 3.1–93.1) vs >35, and female contacts (aOR 2.5 95% CI 1.3–5.0). No contact parameters were associated with acquisition (aOR 1.0–1.1) for any of the duration, frequency, cumulative time in contact, or average duration parameters.

**Conclusions:** We did not find an association between close-range proximity events and SARS-CoV-2 household transmission. Our findings may be due to study limitations, that droplet-mediated transmission during close-proximity contacts plays a smaller role than airborne transmission of SARS-CoV-2 in the household, or due to high contact rates in households.

**Funding:** Wellcome Trust (Grant number 221003/Z/20/Z) in collaboration with the Foreign, Commonwealth, and Development Office, United Kingdom.

## Editor's evaluation

This important study examines the association between SARS-CoV-2 infection and close contact among household members. The authors provide solid evidence that transmission of SARS-CoV-2 within households is not dependent upon close contact. The observations and analyses presented here raise important questions about the mechanics of respiratory pathogen transmission and should inspire future work.

## Introduction

South Africa has experienced five waves of SARS-CoV-2 infection, with over 4 million laboratory-confirmed cases by August 2022 (*National Institute for Communicable Diseases, 2022a*). The true burden is highly underestimated, since based on seroprevalence data, after the third wave of infection, 43 to 83% of the 59.5 million South African inhabitants had already been infected, varying by age and setting (*Kleynhans et al., 2022a*; *Bingham et al., 2022*).

SARS-CoV-2 transmission is mainly via the respiratory route, through both droplet-mediated and airborne transmission (*Meyerowitz et al., 2021*; *Wang et al., 2021*). Infection from contaminated surfaces has also been described (*Meyerowitz et al., 2021*). Although infection risk is highest in symptomatic individuals (*Madewell et al., 2020*), with the most infectious period one day before symptom onset (*Meyerowitz et al., 2021*), asymptomatic individuals can still transmit SARS-CoV-2 (*Liu, 2019*; *Cohen et al., 2022*). Households are a focal point for SARS-CoV-2 transmission (*Aleta et al., 2022*; *Hsu et al., 2021*), especially during peaks of non-pharmaceutical intervention (NPI) restrictions, when movement outside of the household was limited (*Aleta et al., 2022*). Transmission within households can in turn lead to spillover to the community (*Nande et al., 2021*).

Prior to the widespread availability of SARS-CoV-2 vaccines, most countries relied on NPIs to reduce the transmission of the virus, including wearing face masks, social and physical distancing. While mobility and contact survey data showed that the implementation of NPIs led to a reduction in community contacts (*Aleta et al., 2022*; *Liu et al., 2021*) and in turn opportunity for infection, it is still unknown what the role of contact patterns are in the transmission of SARS-CoV-2 in the household. Most analysis relating contact patterns and SARS-CoV-2 transmission done to date has been based on low-resolution data collected from contact tracing (*McAloon et al., 2021*), mobility data (*Aleta et al., 2022*), and contact surveys (*Liu et al., 2021*). To obtain high-resolution contact data, devices broadcasting and receiving radio frequency waves can be used to measure the frequency and duration of close-proximity contacts. This has been used previously to collect contact data in among others, schools (*Salathé et al., 2010*), workplaces (*Cattuto et al., 2010*), hospitals (*Voirin et al., 2015*), and households (*Kiti et al., 2016*), which can, in turn, be used for modeling disease transmission. Specifically, for SARS-CoV-2 so far, high-resolution contact data were collected on cruise ships to identify areas of high contact and to investigate the usefulness of NPIs (*Pung et al., 1956*).

Understanding the drivers of SARS-CoV-2 transmission in the household, especially contact patterns, can help inform NPIs for future SARS-CoV-2 resurgences and potentially future emerging pathogens with pandemic potential. We aimed to assess the association of household close-range contact patterns with the transmission of SARS-CoV-2 in the household using proximity sensors deployed after the identification of SARS-CoV-2 in the household.

## Methods

### Screening, enrolment, and follow-up

We nested a contact study within a case-ascertained, prospective, household transmission study for SARS-CoV-2, implemented in two urban communities in South Africa, Klerksdorp (North West Province) and Soweto (Gauteng Province) from October 2020 through September 2021. Sample size calculations were performed for the main study, but not the nested contact study. For the main study, we aimed to assess a significant difference in the household cumulative infection risk (HCIR) between household contacts exposed to SARS-CoV-2 by a HIV-infected vs HIV-uninfected index case for a 95% confidence interval and 80% power. The resulting total sample size was 440 exposed household members. Detailed sample size calculations and methods for the main study have

been reported previously (*Kleynhans et al., 2022b*). In short, symptomatic adults (aged ≥18 years, symptom onset ≤5 days prior) consulting at clinics were screened for SARS-CoV-2 with real-time reverse transcription polymerase chain reaction (rRT-PCR) on nasopharyngeal swabs. We enrolled household contacts of SARS-CoV-2 infected individuals identified through screening (presumptive index) with ≥2 household contacts (for efficient investigation of risk factors for transmission in the household, weighting cost of household visits and data collected) of whom none reported symptoms prior to index case onset (reducing the probability of previous recent SARS-CoV-2 infection in the household). We visited enrolled households three times a week to collect nasal swabs and data on symptoms and healthcare seeking. At enrolment household characteristics (household size, number of rooms used for sleeping, smoking inside the household, and household income) and individual characteristics (demographics, education, employment, smoking, HIV infection, underlying illness, if SARS-CoV-2 index case was the main caregiver, or sleeping in the same room as index case) were collected. Nasopharyngeal (screening) and nasal swabs (follow-up) were tested for SARS-CoV-2 on rRT-PCR using the Allplex 2019-nCoV kit (Seegene Inc, Seoul, South Korea), and the first positive of each infection episode was characterized using the Allplex SARS-CoV-2 Variants I and II PCR assays (Seegene Inc, Seoul, Korea) and through whole genome sequencing on the Ion Torrent Genexus platform (Thermo Fisher Scientific, USA). We classified the infection episodes as Alpha, Beta, Delta, non-Alpha/Beta/Delta, or unknown variant where we were unable to classify the sample as a variant of concern due to primary testing done elsewhere, low viral load, or poor sequence quality. Households with multiple SARS-CoV-2 variants circulating at the same time (mixed clusters) were excluded from the analysis. We also collected serum at the first and final household visit for serological testing, using an in-house ELISA to detect antibodies against SARS-CoV-2 spike protein (*Wibmer et al., 2021*) and nucleocapsid protein using Roche Elecsys anti-SARS-CoV-2 assay. Individuals were considered sero-positive if they tested positive on either assay. Individuals seropositive at the start of follow-up with no rRT-PCR confirmed SARS-CoV-2 infection during follow-up were excluded from the risk factor analysis for household SARS-CoV-2 acquisition as they may have been protected from infection (*Torresi et al., 2022*), but were still considered in the household size parameter.

## Contact pattern measurements

At the first or second visit during follow-up, we deployed wearable radio frequency (RF) proximity sensors (*Cattuto et al., 2010*) for two weeks to measure close-range interactions (<1.5 meters) between household members. The proximity sensors exchange low-power radio packets in the ISM (Industrial, Scientific, and Medical) radio band. Exchange of packets and Received Signal Strength Indicator (RSSI), suitably thresholded, are used to assess proximity between the devices. A contact interval between two devices is defined as a sequence of consecutive 20 second intervals within which at least one radio packet was exchanged. Each sensor had a unique hardware identifier that was linked to participant study identifiers. Sensors were worn in a PVC pouch either pinned to clothing on the chest, or on a lanyard around the neck based on participant preference. We asked participants to wear the device while at home, to store them separately from other household member sensors at night, and to complete a log sheet every day for the periods the sensors were put on and taken off. During each household visit during the sensor deployment period, field workers confirmed sensors were worn. A deployment log was completed for each household to link the sensor identifier to the participant identifier and to log the date and time sensors were deployed and collected. After sensor collection, batteries were removed to prevent further package exchange between sensors. Sensors were transported to the study office where each sensor was connected to a computer and data downloaded.

## Data analysis

We assumed the first individual with COVID-19-compatible symptoms in the household (individual screened at the clinic) was the index case. Any household member testing positive for SARS-CoV-2 within two weeks from the last positive result for the index case was considered a secondary SARS-CoV-2 case. Contact event data were cleaned using an automated pipeline. We excluded any close-range proximity events outside of the deployment period that occurred during a 5 min time slice that the accelerometer did not detect any movement of the sensor. Accelerometers are very sensitive and even a slight movement will be detected, therefore contacts that occurred while individuals were

sitting/standing still will still be included. Due to a technical error, some sensors at the Klerksdorp site did not have a valid time stamp and needed additional processing to align the time series of close-range proximity events. This was achieved by computing, for each pair of tags X and Y, the temporal shift that maximizes the correlation between the time series of the number of packets per unit time transmitted by X and received by Y, and the reciprocal time series of the number of packets per unit time transmitted by Y and received by X (an operation that can be efficiently carried out working in the frequency domain via Fourier transformation). This allowed us to build a temporal alignment graph between sensors and – as long as there was at least one sensor with a valid timestamp in the household – to use such a graph to propagate the valid timestamp to all other sensors, thus recovering global temporal alignment. For the analysis, we only considered close-range proximity events that occurred one day after sensors were issued and one day before collection, hence excluding any false events logged when sensors were prepared, handed out, and collected in the household. Where no timestamp was available, we used data collected from one to ten days after deployment.

We assessed the following close-range proximity event parameters: (1) median daily duration (median of cumulative duration of close-range proximity events for each day of deployment, in minutes), (2) maximum duration (longest duration of a close-range proximity event during deployment, in minutes), (3) median average daily duration (median of cumulative duration of close-range proximity events in the day divided by the cumulative number of close-range proximity events during that day, in minutes), (4) cumulative time in contact (cumulative duration of close-range proximity events over the deployment period divided by the number of days sensor was worn, in minutes) (5) median daily frequency (median of number of close proximity events for each day of deployment), (6) maximum frequency (highest number of close proximity events in one day during deployment), and (7) daily average frequency (cumulative duration of close-range proximity events over the deployment period divided by the cumulative number of close-range proximity events during the deployment period). Median values were preferred over mean values due to the rightly skewed data, and the different number of days with measured contact data for each household after data cleaning. We assessed contact parameters in two ways: (1) median number of close-range proximity events with the presumptive index case and (2) median number of close-range proximity events with all SARS-CoV-2 infected household members (as confirmed by rRT-PCR). For the group analysis, we did not consider the timing of symptom onset for infected individuals. The latter assessment was to take into account that the transmission could have been from any of the infected household members, and not necessarily the index case, or that the index case was misclassified.

We constructed contact matrices by combining the median duration and frequency of close-range proximity events for all participants between each age group, respectively. To normalize the matrix based on the number of participants, we divided the cumulative contact duration and frequency by the total number of individuals in the two age groups being investigated in each cell.

We assessed the association of contact parameters with SARS-CoV-2 household transmission using the Wilcoxon rank-sum test (considering $p<0.05$ as significant) and through logistic regression controlling for individual characteristics associated with transmission. To assess factors associated with SARS-CoV-2 household transmission, we performed logistic regression with a mixed effects hierarchical regression model to account for household- and site-level clustering. For the analysis with a defined index case (i.e. investigating close-range proximity events with all presumptive index cases, the first person with COVID-19 symptoms), we included only household contacts with their SARS-CoV-2 infection status as the outcome, assessing both index (transmission) and contact (acquisition) characteristics. For the analysis with no defined index case (i.e. investigating close-range proximity events with all SARS-CoV-2 infected household members), we included all enrolled household members (originally considered presumptive index and household contacts), assessing only their own characteristics. For the analysis of close-range proximity events with all SARS-CoV-2 infected household members, we included an offset term in the model to account for the number of SARS-CoV-2 infected members in contact with. We first built the model using individual characteristics to assess factors associated with SARS-CoV-2 transmission (excluding contact parameters). We included age and SARS-CoV-2 variant a priori, and assessed other co-variates on univariate analysis, keeping those with $p<0.2$ in the multivariable analysis. We then performed backward elimination, keeping only those with $p<0.05$, and comparing each subsequent model to the previous using a likelihood ratio test. Finally, we generated a separate model for each close-range proximity parameter, including each parameter in the final

model separately to assess the association with transmission, for both the index and infected household members analysis. As a sensitivity analysis, we also repeated the individual-level analysis restricted to households where no members were excluded due to baseline SARS-CoV-2 seropositivity.

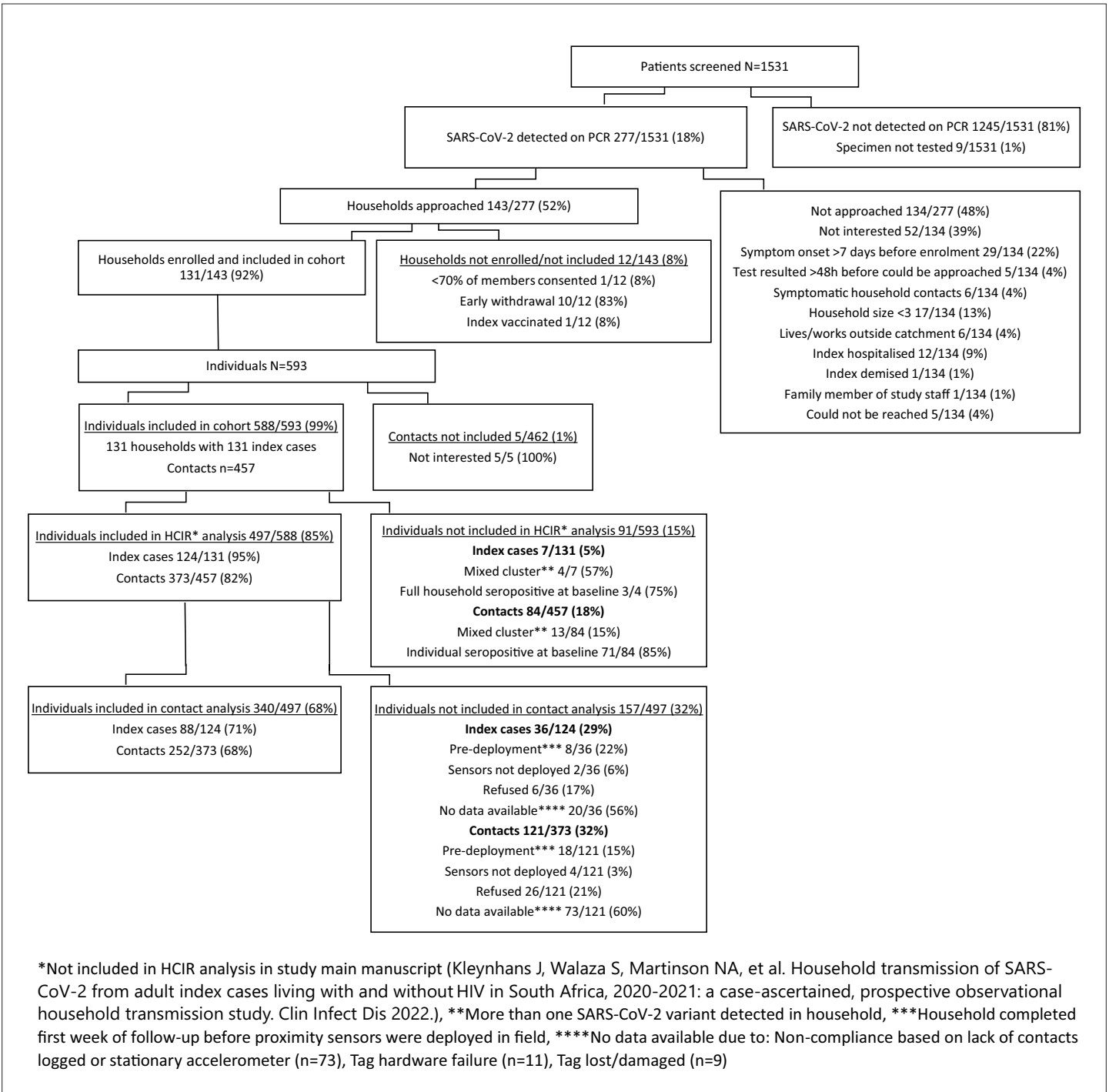

**Figure 1.** Participants screened, enrolled, included in household transmission study, and included in contact analysis, Klerksdorp and Soweto, South Africa, 2020–2021.

## Results

We screened 1531 individuals and identified 277 (18%) positive for SARS-CoV-2, of which 124 (45%) were enrolled and included in the household cumulative infection risk analysis (*Kleynhans et al., 2022b*), with 373 household contacts. After data cleaning, we had contact data for 88 (71%) index cases and 252 (68%) household contacts (*Figure 1*). Ninety-three individuals (19%, 36 index cases, and 73 household contacts) were excluded due to non-compliance, where no contacts were logged, or sensors were stationary for the period based on accelerometer data. We were more likely to have contact data for individuals from the Soweto site, from larger households, and with no household member reporting smoking indoors (*Table 1*). The median number of household members included in the analysis was 4 (interquartile range [IQR] 3–5), with a median of 3 (IQR 1–4) SARS-CoV-2 cases per household and a median of 67% (IQR 50–100%) of household members infected (including index cases). Sixty-six percent (225/340) of individuals included in the analysis lived in a household with 3–5 members, and 49% (168/340) lived in a home with only 1–2 rooms used for sleeping, a third (53/340) living in households where crowding was reported (>2 people per sleeping room *Table 1*).

The overall median daily and maximum duration of close-range proximity events was 18 min (IQR 9–45 min) and 61 min (IQR 25–142 min), respectively. The average duration per close-range proximity event was 0.7 min (IQR 0.5–0.8 min), with a median of 26 (IQR 10–58) close-range proximity events per day amongst household members (*Figures 2 and 3*, *Table 2*). The highest median daily contact duration was observed between individuals within the <5 year, 5–12 year and 35–59 year groups (*Figure 4A and D*). Similar patterns were also seen for median daily close-range proximity duration and frequency in children aged 5–12 and 13–17 years (*Figure 4B–F*).

We did not find any association between any of the contact parameters (either with the index case or all SARS-CoV-2 infected household members) and SARS-CoV-2 infection in the household using the Wilcoxon rank-sum test (p-values ranging from 0.1–0.8, *Table 3*).

When assessing factors associated with SARS-CoV-2 transmission from presumptive index cases and acquisition in household members, none of the contact parameters were associated with SARS-CoV-2 transmission on univariate analysis. Sleeping in the same room as the index case was also not associated with transmission (OR 0.94, 95% CI 0.47–1.88). On multivariable analysis after controlling for index age and SARS-CoV-2 infecting variant, factors significantly associated with higher SARS-CoV-2 transmission and acquisition was index case minimum $C_t$ value <30 (aOR 16.8 95% CI 3.1–93.1) compared to $C_t$ >35, and female contacts (aOR 2.5 95% CI 1.3–5.02). No contact parameters with the index case were associated with acquisition (*Table 4*). Similar results were observed in the sensitivity analysis when households with members seropositive at baseline were excluded (*Table 5*).

When not considering transmission from the presumptive index case, but rather all SARS-CoV-2 infected household members, factors significantly associated with SARS-CoV-2 acquisition on multivariable analysis after controlling for age and SARS-CoV-2 infecting variant were being obese (aOR 4.1 95% CI 1.5–11.1) compared to normal weight, and not currently smoking (aOR 3.2 95% CI 1.2–9.2). No contact parameters with SARS-CoV-2 infected household members were associated with acquisition (*Table 6*).

## Discussion

In this case-ascertained, prospective household transmission study we did not find an association between the duration and frequency of close-range proximity events with SARS-CoV-2 infected household members and transmission in the household.

High-resolution contact patterns have been previously used in the context of pathogen transmission. Examples include investigating influenza virus transmission routes in a hospital setting (*Whitney, 2016*), and contact surveys to show the association between contacts, locations, and influenza infection (*Kwok et al., 2014*). For bacterial infections, the high correlation between pneumococcal infection risk and contact behavior has been shown (*Qian et al., 2022*), and in the context of tuberculosis transmission, it was shown that contact with adults is more important than contact with children (*Dodd et al., 2016*). To our knowledge, there are few data available on the direct association of close-range proximity events and SARS-CoV-2, and none make use of high-resolution contact data. During contact tracing efforts early in the pandemic in Singapore, it was found that sharing a bedroom with an index case and speaking to the index case for 30 min or longer increased the risk for infection (*Ng et al.,*

**Table 1.** Baseline characteristics of SARS-CoV-2 index cases (n=124) and their household contacts (n=373) included in the household cumulative infection risk study and included in the contact study, Klerksdorp and Soweto, South Africa, September 2020–October 2021.

| | Overall | No contact data | Included in contact analysis | p-value |
|---|---|---|---|---|
| | **n=497** | **n=157** | **n=340** | |
| Site | | | | |
| Klerksdorp | 234 (47.1) | 91 (58.0) | 143 (42.1) | 0.001 |
| Soweto | 263 (52.9) | 66 (42.0) | 197 (57.9) | |
| Index | | | | |
| Index | 124 (24.9) | 36 (22.9) | 88 (25.9) | 0.551 |
| Contact | 373 (75.1) | 121 (77.1) | 252 (74.1) | |
| Household size | | | | |
| 3–5 | 347 (69.8) | 122 (77.7) | 225 (66.2) | 0.012 |
| 6–10 | 150 (30.2) | 35 (22.3) | 115 (33.8) | |
| Rooms used for sleeping | | | | |
| 1–2 | 244 (49.1) | 76 (48.4) | 168 (49.4) | 0.387 |
| 3–4 | 203 (40.8) | 69 (43.9) | 134 (39.4) | |
| >4 | 50 (10.1) | 12 (7.6) | 38 (11.2) | |
| Crowding | | | | |
| No | 353 (71.0) | 112 (71.3) | 241 (70.9) | 1 |
| Yes | 144 (29.0) | 45 (28.7) | 99 (29.1) | |
| Child <5 years | | | | |
| No | 423 (85.1) | 136 (86.6) | 287 (84.4) | 0.611 |
| Yes | 74 (14.9) | 21 (13.4) | 53 (15.6) | |
| HH member smokes inside | | | | |
| No | 401 (80.7) | 116 (73.9) | 285 (83.8) | 0.013 |
| Yes | 96 (19.3) | 41 (26.1) | 55 (16.2) | |
| Main water source inside home | | | | |
| No | 350 (70.4) | 120 (76.4) | 230 (67.6) | 0.059 |
| Yes | 147 (29.6) | 37 (23.6) | 110 (32.4) | |
| Main cooking fuel | | | | |
| Electricity | 480 (96.6) | 152 (96.8) | 328 (96.5) | 1 |
| Gas/Paraffin | 17 (3.4) | 5 (3.2) | 12 (3.5) | |
| Monthly household income (US$) | | | | |
| 0–50 | 42 (8.5) | 20 (12.7) | 22 (6.5) | 0.125 |
| 51–100 | 41 (8.2) | 16 (10.2) | 25 (7.4) | |
| 101–190 | 90 (18.1) | 25 (15.9) | 65 (19.1) | |
| 191–375 | 77 (15.5) | 21 (13.4) | 56 (16.5) | |
| 376–750 | 36 (7.2) | 12 (7.6) | 24 (7.1) | |
| >750 | 20 (4.0) | 9 (5.7) | 11 (3.2) | |

*Table 1 continued on next page*

*Table 1 continued*

| | Overall | No contact data | Included in contact analysis | p-value |
|---|---|---|---|---|
| Refused to disclose | 191 (38.4) | 54 (34.4) | 137 (40.3) | |
| **Age (years)** | | | | |
| <5 | 19 (3.8) | 8 (5.1) | 11 (3.2) | 0.711 |
| 5–12 | 73 (14.7) | 25 (15.9) | 48 (14.1) | |
| 13–17 | 60 (12.1) | 18 (11.5) | 42 (12.4) | |
| 18–34 | 130 (26.2) | 45 (28.7) | 85 (25.0) | |
| 35–59 | 163 (32.8) | 47 (29.9) | 116 (34.1) | |
| 60+ | 52 (10.5) | 14 (8.9) | 38 (11.2) | |
| **Sex** | | | | |
| Male | 196 (39.4) | 61 (38.9) | 135 (39.7) | 0.935 |
| Female | 301 (60.6) | 96 (61.1) | 205 (60.3) | |
| **Level of education*** | | | | |
| No schooling/kindergarten | 18 (3.6) | 5 (3.2) | 13 (3.8) | 0.959 |
| Primary | 23 (4.6) | 8 (5.1) | 15 (4.4) | |
| Secondary | 110 (22.1) | 32 (20.4) | 78 (22.9) | |
| Matriculation | 169 (34.0) | 52 (33.1) | 117 (34.4) | |
| Post-secondary | 20 (4.0) | 7 (4.5) | 13 (3.8) | |
| Unknown | 157 (31.6) | 53 (33.8) | 104 (30.6) | |
| **Employment*** | | | | |
| Unemployed | 170 (34.2) | 52 (33.1) | 118 (34.7) | 0.876 |
| Student | 33 (6.6) | 9 (5.7) | 24 (7.1) | |
| Employed | 109 (21.9) | 34 (21.7) | 75 (22.1) | |
| Unknown | 185 (37.2) | 62 (39.5) | 123 (36.2) | |
| **Smoking cigarettes ‡** | | | | |
| No | 65 (13.1) | 22 (14.0) | 43 (12.6) | 0.558 |
| Yes | 426 (85.7) | 132 (84.1) | 294 (86.5) | |
| Unknown | 6 (1.2) | 3 (1.9) | 3 (0.9) | |
| **Living with HIV** | | | | |
| No | 241 (48.5) | 87 (55.4) | 154 (45.3) | 0.095 |
| Yes | 56 (11.3) | 17 (10.8) | 39 (11.5) | |
| Unknown | 200 (40.2) | 53 (33.8) | 147 (43.2) | |
| **Underlying illness†** | | | | |
| No | 416 (83.7) | 128 (81.5) | 288 (84.7) | 0.395 |
| Yes | 71 (14.3) | 24 (15.3) | 47 (13.8) | |
| Unknown | 10 (2.0) | 5 (3.2) | 5 (1.5) | |
| **Body-mass index** | | | | |
| Underweight | 28 (5.6) | 8 (5.1) | 20 (5.9) | 0.757 |
| Normal weight | 207 (41.6) | 67 (42.7) | 140 (41.2) | |

*Table 1 continued on next page*

*Table 1 continued*

|  | Overall | No contact data | Included in contact analysis | p-value |
|---|---|---|---|---|
| Overweight | 100 (20.1) | 31 (19.7) | 69 (20.3) |  |
| Obese | 152 (30.6) | 46 (29.3) | 106 (31.2) |  |
| Unknown | 10 (2.0) | 5 (3.2) | 5 (1.5) |  |
| SARS-CoV-2 infection |  |  |  |  |
| Negative | 153 (30.8) | 54 (34.4) | 99 (29.1) | 0.478 |
| Positive (index) | 124 (24.9) | 36 (22.9) | 88 (25.9) |  |
| Positive (not index) | 220 (44.3) | 67 (42.7) | 153 (45.0) |  |

Values in headers indicate the number of individuals. p-values calculated using the Chi-squared test.

*For individuals ≥18 years old.

†Self-reported history of diabetes, hypertension, asthma, lung disease, heart disease, stroke, spinal cord injury, epilepsy, cancer, liver disease, renal disease, and pre-maturity.

‡For individuals ≥15 years old.

*2021*). We did not see similar results when assessing sharing a bedroom with the index case, and this may be due to the already high level of crowding in included households. Although we observed an increase in infection risk with higher average contact durations with the index case on univariate analysis, this association was no longer seen when adjusting for age and other index and contact factors associated with transmission/acquisition. Mobile device geolocation has also been used to predict contact events between individuals on population level, and was used in transmission models to predict case numbers (*Crawford et al., 2022*). However, we did not find close-range proximity events to be an important driver for household transmission.

There are several possible reasons why we did not observe an association between close-range proximity events and SARS-CoV-2 transmission; these can be classified as related to transmission dynamics or study limitations. One possibility is that along with droplet-mediated transmission during close-proximity contacts, airborne (*Meyerowitz et al., 2021*; *Wang et al., 2021*), and to a lesser extent, fomite-mediated transmission (*Meyerowitz et al., 2021*) may also play a role in the transmission of SARS-CoV-2 in the household. More evidence is becoming available showing that aerosol transmission may be a more important transmission route for SARS-CoV-2 than initially anticipated, especially so in poorly ventilated indoor environments (*Wang et al., 2021*; *Duval et al., 2022*). Households in these communities do not have central air-conditioning or heating (*Mathee et al., 2021*), and during the winter months ventilation may be poorer than in summer, although we did not measure this. Furthermore, sensors only measure face-to-face interactions, and if individuals were close to each other but not directly facing one another for extended durations, we would not have measured this, although sharing of the same air may have occurred. The ventilation within households should be considered in future studies, as this can be a target for intervention strategies to reduce secondary transmission. The high level of interaction in relatively crowded South African households may already be above the threshold for transmission risk, with host characteristics like index viral load and contact age being more important to determine infection risk in this context. It is of interest that close-range proximity patterns within the household did not fully account for the differences in transmission based on age; with teenagers and adults experiencing the highest infection risk, but children aged 5–17 years having the highest contacts.

Our study had limitations both in design and execution. Due to the nature of the case-ascertained study design, we would have missed the period when the index case was most infectious, just before symptom onset (*Meyerowitz et al., 2021*), and the close-range contact patterns measured during the study may have been different after the household members were aware of the index SARS-CoV-2 case (leading to reduced contact), and again once secondary cases were informed of their infection status (leading to increased contact). We also did not collect any information on possible NPI usage in the households, like wearing masks. A study from South Africa showed that individuals staying at home were less likely to wear a mask (*Burger et al., 2022*), but these data were not ascertained

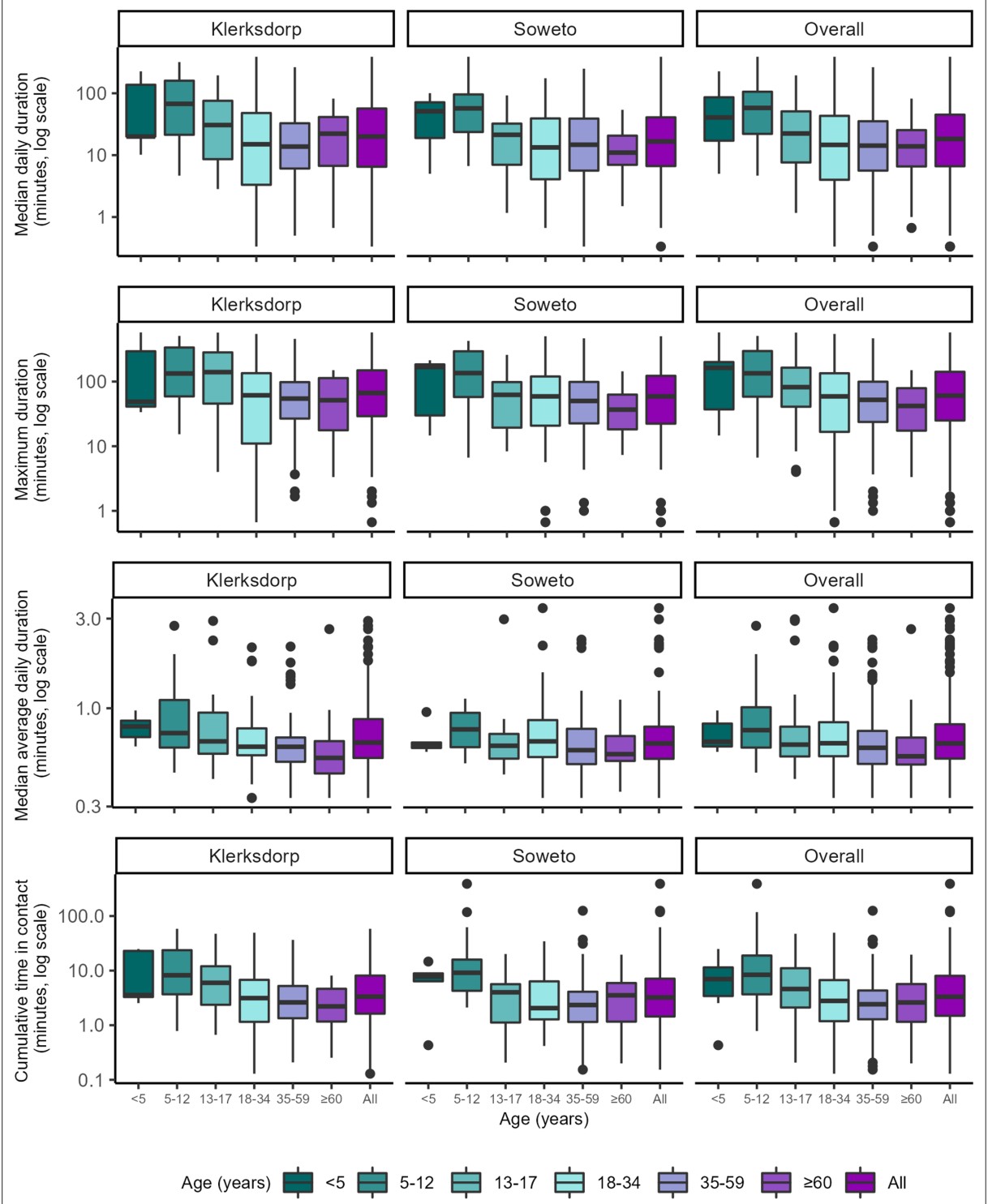

**Figure 2.** Contact parameters related to the duration of all close-range proximity events within households by age group (year) and site, Klerksdorp (n=143) and Soweto (n=197), South Africa, September 2020–October 2021. Median daily duration: median of cumulative duration of close-range proximity events for each day of deployment, in minutes. Maximum duration: longest duration of a close-range proximity event during deployment, in minutes. Median average daily duration: median of cumulative duration of close-range proximity events in the day divided by the cumulative number of close-range proximity events during that day, in minutes. Cumulative time in contact: cumulative duration of close-range proximity events over the deployment period divided by the number of days sensor was worn, in minutes. Horizontal line represents the median, box represents the 25th and 75th percentile, whiskers represent the 1st and 99th percentile, and circles indicate outliers.

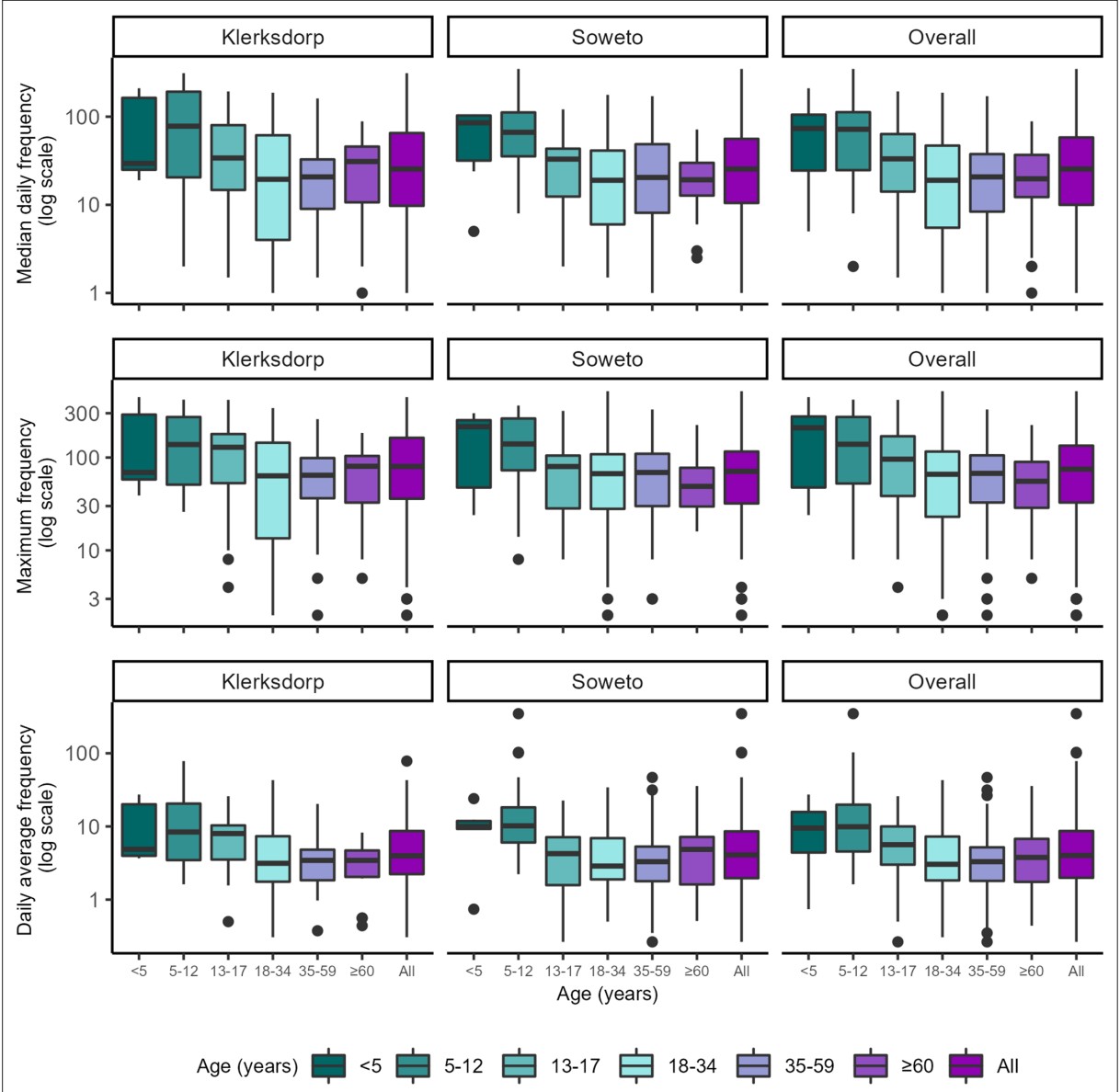

**Figure 3.** Contact parameters related to the frequency of all close-range proximity events within households by age group (year) and site, Klerksdorp (n=143) and Soweto (n=197), South Africa, September 2020–October 2021. Median daily frequency: median of number of close proximity events for each day of deployment. Maximum frequency: highest number of close proximity events in one day during deployment. Daily average frequency: cumulative duration of close-range proximity events over the deployment period divided by the cumulative number of close-range proximity events during the deployment period. Horizontal line represents the median, box represents the 25th and 75th percentile, whiskers represent 1st and 99th percentile, and circles indicate outliers.

during a time when a household member was infected with SARS-CoV-2. We also did not consider where contacts took place (indoors or outdoors), which relates to ventilation and may have influenced transmission. We may have also misclassified the true index case if they were asymptomatic, and did not consider tertiary transmission chains in the index-directed analysis. To adjust for possible misclassification, we performed a grouped assessment investigating close-range proximity events with all SARS-CoV-2 infected household members. This grouped analysis may also have diluted possible associations with the true infector. Furthermore, although based on legislation, close contacts (including household contacts) of SARS-CoV-2 cases were supposed to quarantine, compliance was not monitored. Therefore, household contacts could have been exposed to non-household SARS-CoV-2 cases during the follow-up period. We did not consider multiple introductions of SARS-CoV-2 within the

**Table 2.** Close-range proximity event parameters by age group (year) and site, Klerksdorp and Soweto, South Africa, September 2020–October 2021.

| | n | Median daily duration * | Maximum duration † | Median average daily duration ‡ | Cumulative time in contact (per day) § | Median daily frequency ¶ | Maximum frequency ** | Daily average frequency †† |
|---|---|---|---|---|---|---|---|---|
| **Both sites** | | Median (IQR) | | | | | | |
| Overall | 340 | 18.2 (6.6–45.1) | 60.5 (25.0–141.7) | 0.7 (0.5–0.8) | 3.3 (1.5–8.0) | 25.5 (10.0–58.1) | 75.0 (32.8–134.2) | 4.0 (2.0–8.7) |
| <5 | 11 | 40.7 (17.3–87.2) | 163.0 (37.3–200.8) | 0.7 (0.6–0.8) | 7.0 (3.4–11.8) | 73.5 (24.5–105.2) | 209.0 (48.5–277.0) | 9.5 (4.4–16.2) |
| 5–12 | 48 | 58.0 (22.1–106.3) | 134.3 (58.3–296.7) | 0.8 (0.6–1.0) | 8.4 (3.7–18.9) | 72.2 (24.8–112.5) | 139.0 (52.5–273.5) | 9.9 (4.6–19.8) |
| 13–17 | 42 | 22.4 (7.7–51.2) | 82.0 (40.8–164.1) | 0.6 (0.6–0.8) | 4.6 (2.1–11.1) | 33.2 (14.1–63.5) | 96.0 (38.5–169.5) | 5.6 (3.0–10.0) |
| 18–34 | 85 | 14.7 (4.0–43.2) | 59.0 (16.7–134.3) | 0.7 (0.6–0.8) | 2.8 (1.2–6.7) | 19.0 (5.5–47.0) | 66.0 (23.0–116.0) | 3.0 (1.8–7.3) |
| 35–59 | 116 | 14.2 (5.6–35.3) | 52.2 (23.8–99.5) | 0.6 (0.5–0.8) | 2.4 (1.3–4.3) | 20.8 (8.4–37.6) | 67.5 (32.8–105.5) | 3.3 (1.8–5.2) |
| ≥60 | 38 | 14.0 (6.6–25.4) | 42.0 (17.4–79.6) | 0.6 (0.5–0.7) | 2.6 (1.2–5.7) | 19.8 (12.2–36.9) | 55.5 (28.8–89.8) | 3.8 (1.8–6.8) |
| **Klerksdorp** | | | | | | | | |
| Overall | 143 | 20.0 (6.5–56.8) | 66.7 (29.2–149.2) | 0.7 (0.5–0.9) | 3.3 (1.6–8.1) | 25.5 (9.8–65.2) | 80.0 (36.0–163.0) | 4.0 (2.2–8.7) |
| <5 | 5 | 20.0 (20.0–137.2) | 48.7 (41.0–294.3) | 0.8 (0.7–0.9) | 3.6 (3.3–22.9) | 29.5 (25.0–163.5) | 69.0 (58.0–290.0) | 4.9 (4.0–20.1) |
| 5–12 | 21 | 67.5 (21.3–159.7) | 133.3 (59.0–336.3) | 0.7 (0.6–1.1) | 8.2 (3.7–23.7) | 78.0 (20.5–192.0) | 138.0 (51.0–273.0) | 8.4 (3.5–20.5) |
| 13–17 | 23 | 30.7 (8.8–75.7) | 140.0 (45.8–282.8) | 0.7 (0.6–0.9) | 6.0 (2.4–12.0) | 34.0 (14.8–81.2) | 129.0 (53.0–179.0) | 8.0 (3.5–10.3) |
| 18–34 | 38 | 15.0 (3.5–48.0) | 61.5 (11.0–134.8) | 0.6 (0.6–0.8) | 3.1 (1.2–6.8) | 19.5 (4.0–61.6) | 63.5 (13.5–144.5) | 3.1 (1.8–7.4) |
| 35–59 | 42 | 13.8 (6.1–32.8) | 54.5 (26.8–98.1) | 0.6 (0.5–0.7) | 2.6 (1.3–5.2) | 20.8 (9.0–32.8) | 64.5 (36.5–98.8) | 3.4 (1.8–4.8) |
| ≥60 | 14 | 22.2 (6.8–41.9) | 51.7 (19.8–114.7) | 0.5 (0.5–0.7) | 2.2 (1.2–4.8) | 31.0 (10.8–45.9) | 80.5 (34.0–104.0) | 3.4 (2.1–4.7) |
| **Soweto** | | | | | | | | |
| Overall | 197 | 16.7 (6.7–40.8) | 59.0 (22.3–122.7) | 0.6 (0.5–0.8) | 3.2 (1.4–7.1) | 25.5 (10.5–56.0) | 71.0 (32.0–116.0) | 4.1 (2.0–8.6) |
| <5 | **6 | 52.5 (21.2–71.6) | 167.8 (53.5–184.4) | 0.6 (0.6–0.7) | 7.9 (6.4–8.9) | 86.2 (36.4–103.1) | 215.0 (74.0–253.2) | 9.9 (9.3–11.9) |
| 5–12 | 27 | 57.0 (23.7–95.8) | 135.3 (57.7–294.0) | 0.8 (0.6–0.9) | 9.1 (4.3–16.1) | 66.5 (35.5–111.8) | 140.0 (73.0–263.5) | 10.2 (6.1–18.3) |
| 13–17 | 19 | 21.3 (7.2–32.5) | 62.3 (20.0–98.3) | 0.6 (0.5–0.7) | 4.0 (1.2–5.7) | 33.0 (12.5–43.5) | 80.0 (28.5–105.0) | 4.2 (1.6–7.1) |
| 18–34 | 47 | 13.3 (4.1–39.2) | 59.0 (20.8–120.2) | 0.7 (0.6–0.9) | 2.1 (1.3–6.4) | 19.0 (6.0–41.2) | 67.0 (28.0–108.5) | 2.9 (1.9–6.9) |
| 35–59 | 74 | 14.8 (5.6–39.0) | 50.2 (22.6–98.8) | 0.6 (0.5–0.8) | 2.3 (1.1–4.1) | 20.5 (8.1–49.1) | 69.0 (30.0–109.8) | 3.3 (1.8–5.3) |
| ≥60 | 24 | 11.0 (7.0–20.7) | 37.0 (18.2–62.8) | 0.6 (0.5–0.7) | 3.6 (1.2–5.9) | 19.2 (12.8–30.0) | 49.0 (29.8–77.8) | 4.9 (1.6–7.2) |

*Median daily duration (median of cumulative duration of close-range proximity events for each day of deployment, in minutes).

†Maximum duration (longest duration of a close-range proximity event during deployment, in minutes).

‡Median average daily duration (median of cumulative duration of close-range proximity events in the day divided by the cumulative number of close-range proximity events during that day, in minutes).

§Cumulative time in contact (cumulative duration of close-range proximity events over the deployment period divided by the number of days sensor was worn, in minutes).

¶Median daily frequency (median of number of close proximity events for each day of deployment).

**Maximum frequency (highest number of close proximity events in one day during deployment).

††Daily average frequency (cumulative duration of close-range proximity events over the deployment period divided by the cumulative number of close-range proximity events during the deployment period).

household, although we did exclude households with more than one SARS-CoV-2 variant detected. During the peaks of waves of infection in South Africa, one variant was responsible for the majority of the infections (**National Institute for Communicable Diseases, 2022b**), and the additional introductions within the household were likely to have been the same as the initial variant. Higher resolution sequencing data may be useful to more accurately identify chains of transmission within the household. Combining contact data with clinical and virological/bacteriological data has been shown to be useful to reconstruct transmission networks (**Campbell et al., 2019**), and we will consider this for

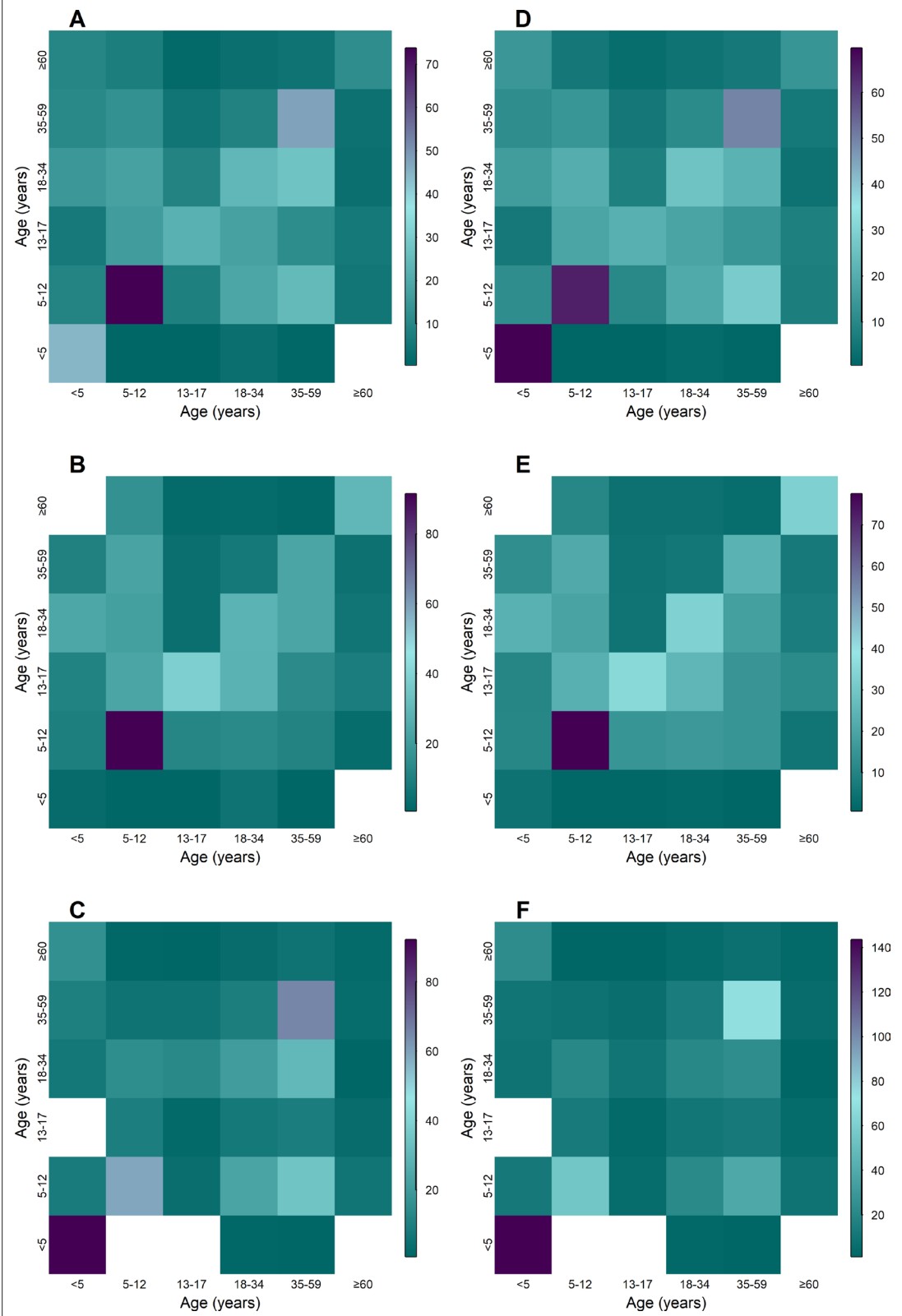

**Figure 4.** Aged-based contact matrices based close-proximity event duration (**A–C**) and frequency (**D–E**) for the entire deployment period overall (**A, D**) Klerksdorp (**B,D**, n=143), and Soweto (**C,F**, n=193), September 2020–October 2021. Teal denotes the lowest value, purple highest, and white no data for age group combination.

**Table 3.** Association of contact parameters with SARS-CoV-2 household acquisition * using the Wilcoxon rank-sum test, Klerksdorp and Soweto, South Africa, September 2020–October 2021.

| Contact parameter | p-value (including all households) |
|---|---|
| Median daily duration with index [†] | 0.83 |
| Maximum duration with index [‡] | 0.32 |
| Median average daily duration with index [§] | 0.78 |
| Cumulative time in contact with index [¶] | 0.83 |
| Median daily frequency with index [**] | 0.71 |
| Maximum frequency with index [††] | 0.57 |
| Daily average frequency with index [‡‡] | 0.54 |
| Median daily duration with infected household members [†] | 0.25 |
| Maximum duration with infected household members [‡] | 0.79 |
| Median average daily duration with infected household members [§] | 0.27 |
| Cumulative time in contact with infected household members [¶] | 0.14 |
| Median daily frequency with infected household members [**] | 0.32 |
| Maximum frequency with infected household members [††] | 0.76 |
| Daily average frequency with infected household members [‡‡] | 0.18 |

*Outcome investigated: testing positive for SARS-CoV-2.

[†]Median daily duration (median of cumulative duration of close-range proximity events for each day of deployment, in minutes).

[‡]Maximum duration (longest duration of a close-range proximity event during deployment, in minutes).

[§]Median average daily duration (median of cumulative duration of close-range proximity events in the day divided by the cumulative number of close-range proximity events during that day, in minutes).

[¶]Cumulative time in contact (cumulative duration of close-range proximity events over the deployment period divided by the number of days sensor was worn, in minutes).

[**]Median daily frequency (median of number of close proximity events for each day of deployment).

[††]Maximum frequency (highest number of close proximity events in one day during deployment).

[‡‡]Daily average frequency (cumulative duration of close-range proximity events over the deployment period divided by the cumulative number of close-range proximity events during the deployment period).

future analyses. Our measurement of close-range contact patterns was also limited by compliance, as during the cleaning process we identified 73 sensors that were not worn, based on accelerometer data. We also had limited data in some households, where some individuals did not consent to the contact aspect of the study, or where we were unable to retrieve data due to hardware failure, lost, or damaged tags. The small sample size may have reduced our power to detect small differences in close proximity event parameters between those infected with SARS-CoV-2 and those not infected.

In conclusion, we did not observe an association between close-proximity contacts and SARS-CoV-2 transmission in the household. A case-ascertained, prospective household transmission study may not be well suited to investigate this question. A possible other study design to consider is randomly selected prospective household cohorts, but the deployment of sensors for extended periods of time may be logistically challenging and lead to participant fatigue, and households in a cohort may not experience infection episodes unless the community attack rate is very high. High-resolution contacts in other settings like schools or workplaces where contacts are less frequent could be useful to identify the type of contact events that may lead to SARS-CoV-2 transmission. If aerosol transmission plays a more important role in transmission than droplet-mediated transmission, ventilation within households can also be an important consideration for future studies. Increased ventilation could potentially be a method to reduce secondary transmission in households. Nevertheless, our study provides high-resolution household contact data that can be used to parametrize future transmission models, not only for SARS-CoV-2, but other pathogens as well.

**Table 4.** Factors associated with SARS-CoV-2 household transmission from index cases and acquisition in household contacts (contact parameters with index case), Klerksdorp and Soweto, South Africa, 2020–2021 (n=252).

| | SARS-CoV-2 infection* | | Univariate analysis | Multivariable analysis | Multivariable analysis (including contact parameter) | | | | | | |
| --- | --- | --- | --- | --- | --- | --- | --- | --- | --- | --- | --- |
| | Negative n=99 | Positive n=153 | OR (95% CI) | aOR (95% CI) | aOR (95% CI) | aOR (95% CI) | aOR (95% CI) | aOR (95% CI) | aOR (95% CI) | aOR (95% CI) | aOR (95% CI) |
| **Index Characteristics** | | | | | | | | | | | |
| **Site** | | | | | | | | | | | |
| Klerksdorp | 47/106 (44%) | 59/106 (56%) | Reference | | | | | | | | |
| Soweto | 52/146 (36%) | 94/146 (64%) | 1.73 (0.72–4.14) | | | | | | | | |
| **Age (years)** | | | | | | | | | | | |
| 18–34 | 36/76 (47%) | 40/76 (53%) | Reference | Reference | Reference | Reference | Reference | Reference | Reference | Reference | Reference |
| 35–59 | 48/137 (35%) | 89/137 (65%) | 2.20 (0.80–6.14) | 2.21 (0.80–6.02) | 2.24 (0.79–6.32) | 2.17 (0.77–6.14) | 2.21 (0.79–6.15) | 2.51 (0.85–7.40) | 2.43 (0.85–6.94) | 2.19 (0.77–6.23) | 2.64 (0.89–7.85) |
| ≥60 | 15/39 (38%) | 24/39 (62%) | 1.86 (0.47–7.37) | 2.08 (0.51–8.50) | 2.10 (0.50–8.82) | 1.96 (0.47–8.25) | 2.06 (0.49–8.57) | 2.40 (0.55–10.49) | 2.34 (0.55–9.98) | 2.03 (0.48–8.59) | 2.43 (0.55–10.80) |
| **Minimum C_t value** | | | | | | | | | | | |
| >35 | 17/21 (81%) | 4/21 (19%) | Reference | Reference | Reference | Reference | Reference | Reference | Reference | Reference | Reference |
| 30–35 | 23/50 (46%) | 27/50 (54%) | 7.58 (1.21–47.43) | 7.07 (0.75–66.23) | 7.23 (0.76–68.84) | 6.83 (0.72–64.85) | 6.88 (0.72–65.28) | 7.98 (0.79–80.47) | 7.62 (0.78–74.23) | 6.59 (0.68–64.07) | 8.06 (0.78–83.67) |
| <30 | 58/178 (33%) | 120/178 (67%) | 16.84 (3.05–93.06) | 10.60 (1.40–80.08) | 10.77 (1.41–82.26) | 10.39 (1.36–79.27) | 10.36 (1.36–79.23) | 11.04 (1.38–88.37) | 11.08 (1.42–86.47) | 10.04 (1.29–78.05) | 11.05 (1.35–90.60) |
| Unknown | 1/3 (33%) | 2/3 (67%) | 15.69 (0.31–793.51) | 7.22 (0.12–438.47) | 7.27 (0.12–450.06) | 6.79 (0.11–423.55) | 7.37 (0.12–465.42) | 7.55 (0.11–503.58) | 7.42 (0.12–467.86) | 6.85 (0.10–453.16) | 7.92 (0.11–558.92) |
| **SARS-CoV-2 variant** | | | | | | | | | | | |
| non-Alpha/Beta/Delta | 5/14 (36%) | 9/14 (64%) | 1.22 (0.22–6.93) | 2.04 (0.31–13.41) | 2.12 (0.32–14.08) | 2.15 (0.32–14.51) | 2.06 (0.31–13.72) | 2.00 (0.29–13.76) | 1.99 (0.30–13.26) | 1.99 (0.29–13.61) | 2.07 (0.29–14.67) |
| Alpha | 2/13 (15%) | 11/13 (85%) | 4.81 (0.59–39.19) | 5.02 (0.52–48.39) | 5.15 (0.48–55.49) | 5.48 (0.53–57.15) | 4.71 (0.43–51.12) | 3.45 (0.33–36.67) | 4.59 (0.47–44.90) | 5.02 (0.50–50.59) | 4.91 (0.47–51.69) |
| Beta | 70/171 (41%) | 101/171 (59%) | Reference | Reference | Reference | Reference | Reference | Reference | Reference | Reference | Reference |
| Delta | 8/38 (21%) | 30/38 (79%) | 3.24 (0.93–11.32) | 3.76 (0.97–14.55) | 3.79 (0.96–14.95) | 3.81 (0.97–14.89) | 3.90 (0.99–15.29) | 3.86 (0.96–15.53) | 3.88 (0.99–15.27) | 3.87 (0.97–15.53) | 3.94 (0.96–16.23) |
| Variant Unknown | 14/16 (88%) | 2/16 (12%) | 0.06 (0.01–0.41) | 0.15 (0.01–1.60) | 0.16 (0.02–1.67) | 0.16 (0.01–1.66) | 0.15 (0.01–1.56) | 0.16 (0.01–1.56) | 0.14 (0.01–1.50) | 0.14 (0.01–1.55) | 0.14 (0.01–1.61) |
| **Contact characteristics** | | | | | | | | | | | |
| **Age (years)** | | | | | | | | | | | |
| <5 | 8/11 (73%) | 3/11 (27%) | 0.17 (0.03–1.05) | 0.28 (0.04–1.90) | 0.30 (0.04–2.04) | 0.31 (0.05–2.11) | 0.28 (0.04–1.90) | 0.24 (0.03–1.70) | 0.24 (0.03–1.97) | 0.27 (0.04–1.97) | 0.22 (0.03–1.59) |
| 5–12 | 21/48 (44%) | 27/48 (56%) | 0.69 (0.25–1.93) | 0.54 (0.19–1.52) | 0.53 (0.18–1.54) | 0.55 (0.19–1.56) | 0.51 (0.18–1.46) | 0.49 (0.17–1.42) | 0.47 (0.16–1.37) | 0.51 (0.17–1.48) | 0.48 (0.17–1.41) |
| 13–17 | 11/42 (26%) | 31/42 (74%) | 2.55 (0.78–8.32) | 2.41 (0.72–8.08) | 2.42 (0.72–8.16) | 2.42 (0.72–8.17) | 2.38 (0.70–8.03) | 2.47 (0.72–8.45) | 2.46 (0.73–8.34) | 2.39 (0.71–8.12) | 2.57 (0.75–8.83) |
| 18–34 | 22/60 (37%) | 38/60 (63%) | Reference | Reference | Reference | Reference | Reference | Reference | Reference | Reference | Reference |
| 35–59 | 27/68 (40%) | 41/68 (60%) | 0.72 (0.28–1.87) | 0.60 (0.23–1.57) | 0.60 (0.23–1.59) | 0.61 (0.23–1.61) | 0.59 (0.22–1.56) | 0.60 (0.23–1.59) | 0.59 (0.22–1.57) | 0.58 (0.22–1.55) | 0.62 (0.23–1.65) |
| ≥60 | 10/23 (43%) | 13/23 (57%) | 0.78 (0.21–2.93) | 0.61 (0.17–2.24) | 0.61 (0.17–2.27) | 0.61 (0.17–2.27) | 0.61 (0.16–2.23) | 0.65 (0.17–2.46) | 0.64 (0.17–2.37) | 0.61 (0.16–2.26) | 0.66 (0.17–2.54) |
| **Sex** | | | | | | | | | | | |
| Male | 52/108 (48%) | 56/108 (52%) | Reference | Reference | Reference | Reference | Reference | Reference | Reference | Reference | Reference |
| Female | 47/144 (33%) | 97/144 (67%) | 2.51 (1.25–5.02) | 2.38 (1.17–4.84) | 2.37 (1.15–4.86) | 2.44 (1.18–5.03) | 2.32 (1.14–4.74) | 2.25 (1.09–4.63) | 2.20 (1.07–4.53) | 2.36 (1.14–4.88) | 2.21 (1.07–4.59) |

*Table 4 continued on next page*

*Table 4 continued*

| | SARS-CoV-2 infection* | | Univariate analysis | Multivariable analysis | Multivariable analysis (including contact parameter) |
|---|---|---|---|---|---|
| **Sleep in same room as index** | | | | | |
| No | 68/171 (40%) | 103/171 (60%) | Reference | | |
| Yes | 31/81 (38%) | 50/81 (62%) | 0.94 (0.47–1.88) | | |
| **Cared for by index** | | | | | |
| No | 84/212 (40%) | 128/212 (60%) | Reference | | |
| Yes | 15/40 (38%) | 25/40 (62%) | 0.92 (0.36–2.34) | | |
| Median daily duration † | 5 (1–11) | 4 (2–12) | 0.99 (0.98–1.01) | 1.00 (0.98–1.02) | |
| Maximum duration ‡ | 17 (5–48) | 14 (4–38) | 1.00 (0.99–1.00) | 1.00 (0.99–1.01) | |
| Median average daily duration § | 0.53 (0.43–0.69) | 0.56 (0.44–0.67) | 1.15 (0.60–2.21) | 1.08 (0.50–2.30) | |
| Cumulative time in contact ¶ | 0.87 (0.37–1.80) | 0.90 (0.33–2.27) | 1.04 (0.92–1.17) | | 1.08 (0.94–1.25) |
| Median daily frequency ¶ | 6 (3–15) | 7 (3–15) | 1.00 (0.98–1.02) | 1.01 (0.99–1.03) | |
| Maximum frequency ** | 22 (8–52) | 19 (7–41) | 1.00 (0.99–1.01) | 1.00 (0.99–1.01) | |
| Daily average frequency †† | 1.00 (0.57–2.77) | 1.16 (0.53–2.70) | 1.06 (0.95–1.18) | 1.10 (0.96–1.25) | |

\* n/ row N (%); Median (interquartile range).

†Median daily duration (median of cumulative duration of close-range proximity events for each day of deployment, in minutes).

‡Maximum duration (longest duration of a close-range proximity event during deployment, in minutes).

§Median average daily duration (median of cumulative duration of close-range proximity events in the day divided by the cumulative number of close-range proximity events during that day, in minutes).

¶Cumulative time in contact (cumulative duration of close-range proximity events over the deployment period divided by the number of days sensor was worn, in minutes).

\*\*Maximum frequency (highest number of close proximity events in one day during deployment).

††Daily average frequency (cumulative duration of close-range proximity events over the deployment period divided by the cumulative number of close-range proximity events during the deployment period). aOR: adjusted odds ratio. Significant associations on multivariable analysis in boldface. Factors investigated but not found significant on multivariable analysis: index sex, HIV status, underlying conditions, body mass index, current smoking, episode duration, serostatus at follow-up end; contact HIV status, underlying conditions, body mass index, current smoking, cared for by index.

**Table 5.** Factors associated with SARS-CoV-2 household transmission from index cases and acquisition in household contacts (contact parameters with index case) in households with no members excluded from analysis, Klerksdorp and Soweto, South Africa, 2020–2021, (n=192).

| | SARS-CoV-2 infection* | | Univariate analysis | Multivariable analysis | | | | Multivariable analysis (including contact parameter) | | | |
|---|---|---|---|---|---|---|---|---|---|---|---|
| | Negative n=66 | Positive n=126 | OR (95% CI) | aOR (95% CI) | | | | aOR (95% CI) | | | |
| **Index Characteristics** | | | | | | | | | | | |
| **Site** | | | | | | | | | | | |
| Klerksdorp | 25/70 (36%) | 45/70 (64%) | Reference | | | | | | | | |
| Soweto | 41/122 (34%) | 81/122 (66%) | 1.30 (0.42–4.00) | | | | | | | | |
| **Age (years)** | | | | | | | | | | | |
| 18–34 | 20/48 (42%) | 28/48 (58%) | Reference | Reference | Reference | Reference | Reference | Reference | Reference | Reference | Reference |
| 35–59 | 32/111 (29%) | 79/111 (71%) | 2.52 (0.70–9.13) | 2.13 (0.60–7.55) | 1.99 (0.60–6.55) | 1.88 (0.58–6.15) | 2.09 (0.63–6.97) | 1.98 (0.55–7.11) | 2.02 (0.59–6.89) | 1.86 (0.54–6.38) | 2.09 (0.63–7.00) |
| ≥60 | 14/33 (42%) | 19/33 (58%) | 1.01 (0.20–5.19) | 1.41 (0.24–8.22) | 1.31 (0.24–7.23) | 1.22 (0.23–6.62) | 1.38 (0.24–7.84) | 1.34 (0.23–7.74) | 1.35 (0.24–7.72) | 1.26 (0.23–6.94) | 1.43 (0.26–7.91) |
| **Minimum Ct value** | | | | | | | | | | | |
| >35 | 16/18 (89%) | 2/18 (11%) | Reference | Reference | Reference | Reference | Reference | Reference | Reference | Reference | Reference |
| 30–35 | 15/38 (39%) | 23/38 (61%) | 22.18 (2.39–205.39) | 23.38 (1.23–445.33) | 22.55 (1.43–354.91) | 21.38 (1.41–325.01) | 23.36 (1.40–389.09) | 21.73 (1.24–381.17) | 21.25 (1.30–347.22) | 18.86 (1.18–301.62) | 22.49 (1.37–368.46) |
| <30 | 34/133 (26%) | 99/133 (74%) | 48.47 (5.80–404.75) | 39.72 (2.69–585.90) | 38.32 (3.24–452.80) | 37.61 (3.28–431.72) | 39.80 (3.20–495.62) | 37.67 (2.76–513.85) | 36.87 (3.03–448.62) | 32.87 (2.61–413.51) | 38.58 (3.14–473.57) |
| Unknown | 1/3 (33%) | 2/3 (67%) | 30.11 (0.61–1,497.22) | 14.55 (0.18–1,148.81) | 14.11 (0.22–923.86) | 13.46 (0.22–836.17) | 15.49 (0.22–1,110.09) | 14.21 (0.20–1,025.63) | 13.14 (0.19–918.99) | 11.86 (0.19–755.20) | 13.83 (0.20–952.08) |
| **SARS-CoV-2 variant** | | | | | | | | | | | |
| non-Alpha/Beta/Delta | 3/8 (38%) | 5/8 (62%) | 0.68 (0.09–5.16) | 0.68 (0.09–5.16) | 1.71 (0.14–20.48) | 1.79 (0.13–24.54) | 1.76 (0.14–21.84) | 1.85 (0.14–22.72) | 1.75 (0.14–22.72) | 1.71 (0.13–21.95) | 1.60 (0.13–20.06) |
| Alpha | 1/11 (9.1%) | 10/11 (91%) | 7.00 (0.47–103.30) | 7.00 (0.47–103.30) | 8.18 (0.32–206.89) | 5.58 (0.32–96.65) | 6.52 (0.32–132.55) | 5.76 (0.25–133.53) | 7.47 (0.29–194.23) | 5.51 (0.29–103.13) | 5.83 (0.36–94.07) |

*Table 5 continued on next page*

*Table 5 continued*

| | SARS-CoV-2 infection* | | Univariate analysis | Multivariable analysis | Multivariable analysis (including contact parameter) | | | | | | |
|---|---|---|---|---|---|---|---|---|---|---|---|
| **Beta** | 45/130 (35%) | 85/130 (65%) | Reference | Reference | Reference | Reference | Reference | Reference | Reference | Reference | Reference |
| Delta | 7/33 (21%) | 26/33 (79%) | 2.17 (0.61–7.64) | 2.17 (0.61–7.64) | 3.01 (0.67–13.53) | 2.82 (0.68–11.69) | 2.90 (0.68–12.35) | 3.01 (0.69–13.17) | 2.91 (0.68–12.50) | 2.85 (0.66–12.41) | 2.73 (0.65–11.51) |
| Variant Unknown | 10/10 (100%) | 0/10 (0%) | NA | NA | NA | NA | NA | NA | NA | NA | NA |
| **Contact characteristics** | | | | | | | | | | | |
| **Age (years)** | | | | | | | | | | | |
| <5 | 7/8 (88%) | 1/8 (12%) | 0.01 (0.00–0.27) | 0.03 (0.00–0.50) | 0.03 (0.00–0.51) | 0.03 (0.00–0.48) | 0.03 (0.00–0.46) | 0.03 (0.00–0.52) | 0.03 (0.00–0.48) | 0.03 (0.00–0.65) | 0.03 (0.00–0.44) |
| 5–12 | 14/37 (38%) | 23/37 (62%) | 0.38 (0.10–1.40) | 0.33 (0.09–1.20) | 0.38 (0.10–1.34) | 0.35 (0.10–1.23) | 0.33 (0.10–1.16) | 0.34 (0.09–1.23) | 0.35 (0.10–1.24) | 0.37 (0.10–1.37) | 0.33 (0.09–1.14) |
| 13–17 | 7/33 (21%) | 26/33 (79%) | 1.68 (0.38–7.41) | 1.66 (0.34–8.03) | 1.57 (0.33–7.58) | 1.63 (0.33–7.96) | 1.65 (0.33–8.22) | 1.61 (0.34–7.72) | 1.68 (0.34–8.31) | 1.63 (0.35–7.65) | 1.66 (0.33–8.25) |
| 18–34 | 10/42 (24%) | 32/42 (76%) | Reference | Reference | Reference | Reference | Reference | Reference | Reference | Reference | Reference |
| 35–59 | 23/57 (40%) | 34/57 (60%) | 0.36 (0.11–1.19) | 0.35 (0.11–1.13) | 0.36 (0.11–1.10) | 0.35 (0.11–1.10) | 0.35 (0.11–1.09) | 0.34 (0.10–1.09) | 0.36 (0.11–1.11) | 0.36 (0.11–1.16) | 0.34 (0.11–1.08) |
| ≥60 | 5/15 (33%) | 10/15 (67%) | 0.47 (0.08–2.68) | 0.42 (0.08–2.30) | 0.41 (0.08–2.09) | 0.42 (0.08–2.16) | 0.43 (0.08–2.22) | 0.41 (0.08–2.19) | 0.43 (0.08–2.23) | 0.41 (0.08–2.19) | 0.41 (0.08–2.12) |
| **Sex** | | | | | | | | | | | |
| Male | 33/79 (42%) | 46/79 (58%) | Reference | Reference | Reference | Reference | Reference | Reference | Reference | Reference | Reference |
| Female | 33/113 (29%) | 80/113 (71%) | **2.40 (1.05–5.46)** | 2.28 (0.96–5.43) | **2.42 (1.04–5.66)** | **2.38 (1.01–5.62)** | 2.30 (0.99–5.36) | 2.36 (0.99–5.63) | 2.34 (0.99–5.55) | **2.45 (1.02–5.87)** | 2.29 (0.97–5.39) |
| **Sleep in same room as index** | | | | | | | | | | | |
| No | 46/133 (35%) | 87/133 (65%) | Reference | | | | | | | | |
| Yes | 20/59 (34%) | 39/59 (66%) | 0.90 (0.39–2.08) | | | | | | | | |
| **Cared for by index** | | | | | | | | | | | |
| No | 54/160 (34%) | 106/160 (66%) | Reference | | | | | | | | |
| Yes | 12/32 (38%) | 20/32 (62%) | 0.60 (0.19–1.86) | | | | | | | | |

*Table 5 continued*

| | SARS-CoV-2 infection* | Univariate analysis | Multivariable analysis | Multivariable analysis (including contact parameter) |
|---|---|---|---|---|
| Median daily duration † | 6 (1-14) | 4 (1-12) | 0.99 (0.97–1.01) | 0.99 (0.97–1.02) |
| Maximum duration ‡ | 18 (5-51) | 13 (3-37) | 1.00 (0.99–1.00) | 1.00 (0.99–1.01) |
| Median average daily duration § | 0.53 (0.43–0.75) | 0.52 (0.42–0.67) | 1.10 (0.53–2.27) | 1.00 (0.42–2.34) |
| Cumulative time in contact ¶ | 1.01 (0.37–2.26) | 0.88 (0.32–2.34) | 0.98 (0.85–1.12) | 0.96 (0.80–1.16) |
| Median daily frequency ** | 9 (3-16) | 7 (3-15) | 0.99 (0.97–1.01) | 1.00 (0.97–1.02) |
| Maximum frequency †† | 24 (8-68) | 19 (6-40) | 0.99 (0.98–1.00) | 0.99 (0.98–1.01) |
| Daily average frequency ‡‡ | 1.40 (0.60–3.15) | 1.07 (0.50–2.70) | 0.98 (0.84–1.15) | 0.98 (0.81–1.18) |

* n/ row N (%); Median (interquartile range).

†Median daily duration (median of cumulative duration of close-range proximity events for each day of deployment, in minutes).

‡Maximum duration (longest duration of a close-range proximity event during deployment, in minutes).

§Median average daily duration (median of cumulative duration of close-range proximity events in the day divided by the cumulative number of close-range proximity events during that day, in minutes).

¶Cumulative time in contact (cumulative duration of close-range proximity events over the deployment period divided by the number of days sensor was worn, in minutes).

**Median daily frequency (median of number of close proximity events for each day of deployment).

††Maximum frequency (highest number of close proximity events in one day during deployment).

‡‡Daily average frequency (cumulative duration of close-range proximity events over the deployment period divided by the cumulative number of close-range proximity events during the deployment period). aOR: adjusted odds ratio. Significant associations on multivariable analysis in boldface.

**Table 6.** Factors associated with SARS-CoV-2 acquisition within the household (contact parameters with SARS-CoV-2 infected household members), Klerksdorp and Soweto, South Africa, 2020–2021, (n=340).

| | SARS-CoV-2 infection | | Univariate analysis | Multivariable analysis | Multivariable analysis (including contact parameter) | | | | | |
|---|---|---|---|---|---|---|---|---|---|---|
| | Negative n=99 | Positive n=241 | OR (95% CI) | aOR (95% CI) | aOR (95% CI) | aOR (95% CI) | aOR (95% CI) | aOR (95% CI) | aOR (95% CI) | aOR (95% CI) |
| **Site** | | | | | | | | | | |
| Klerksdorp | 47/143 (33%) | 96/143 (67%) | Reference | | | | | | | |
| Soweto | 52/197 (26%) | 145/197 (74%) | 1.70 (0.64–4.51) | | | | | | | |
| **Contact Age (years)** | | | | | | | | | | |
| <5 | 8/11 (73%) | 3/11 (27%) | 0.06 (0.01–0.41) | 1.88 (0.25–14.08) | 0.09 (0.01–0.71) | 0.12 (0.01–0.96) | 0.11 (0.01–0.89) | 0.09 (0.01–0.75) | 0.15 (0.02–1.21) | 0.08 (0.01–0.63) |
| 5–12 | 21/48 (44%) | 27/48 (56%) | 0.20 (0.07–0.57) | 9.43 (1.17–75.88) | 0.17 (0.05–0.56) | 0.24 (0.08–0.75) | 0.21 (0.07–0.62) | 0.18 (0.05–0.59) | 0.25 (0.08–0.76) | 0.15 (0.05–0.48) |
| 13–17 | 11/42 (26%) | 31/42 (74%) | 0.87 (0.28–2.70) | 10.08 (1.34–76.04) | 0.96 (0.28–3.28) | 1.00 (0.30–3.36) | 1.12 (0.34–3.74) | 0.95 (0.28–3.25) | 1.05 (0.32–3.44) | 0.92 (0.28–3.04) |
| 18–34 | 22/85 (26%) | 63/85 (74%) | Reference | Reference | Reference | Reference | Reference | Reference | Reference | Reference |
| 35–59 | 27/116 (23%) | 89/116 (77%) | 0.98 (0.41–2.30) | 8.79 (1.14–67.74) | 0.87 (0.34–2.23) | 0.91 (0.36–2.30) | 0.84 (0.34–2.12) | 0.88 (0.34–2.24) | 0.92 (0.37–2.27) | 0.87 (0.35–2.15) |
| ≥60 | 10/38 (26%) | 28/38 (74%) | 1.11 (0.34–3.58) | 8.71 (0.98–77.54) | 0.86 (0.24–3.05) | 0.86 (0.25–3.00) | 0.85 (0.25–2.87) | 0.87 (0.25–3.05) | 0.89 (0.27–2.97) | 0.85 (0.25–2.88) |
| **Contact Sex** | | | | | | | | | | |
| Male | 52/135 (39%) | 83/135 (61%) | Reference | | | | | | | |
| Female | 47/205 (23%) | 158/205 (77%) | 2.64 (1.40–4.95) | | | | | | | |
| **Body mass index** | | | | | | | | | | |
| Normal weight | 50/140 (36%) | 90/140 (64%) | Reference | Reference | Reference | Reference | Reference | Reference | Reference | Reference |
| Underweight | 7/20 (35%) | 13/20 (65%) | 0.89 (0.22–3.53) | 0.90 (0.22–3.67) | 0.87 (0.20–3.87) | 0.88 (0.21–3.73) | 0.78 (0.19–3.29) | 0.90 (0.20–3.96) | 1.01 (0.25–4.09) | 0.81 (0.19–3.38) |
| Overweight | 22/69 (32%) | 47/69 (68%) | 1.61 (0.70–3.71) | 1.17 (0.49–2.76) | 1.17 (0.47–2.91) | 1.17 (0.49–2.78) | 1.18 (0.49–2.82) | 1.16 (0.47–2.89) | 1.14 (0.48–2.67) | 1.17 (0.49–2.80) |
| Obese | 17/106 (16%) | 89/106 (84%) | 7.47 (3.10–17.98) | 4.14 (1.54–11.11) | 4.31 (1.54–12.03) | 3.83 (1.40–10.48) | 4.40 (1.62–11.92) | 4.20 (1.51–11.69) | 3.84 (1.44–10.23) | 4.33 (1.60–11.69) |
| Unknown | 3/5 (60%) | 2/5 (40%) | 0.53 (0.03–10.58) | NA | NA | NA | NA | NA | NA | NA |
| **Current smoking** | | | | | | | | | | |
| Yes | 18/43 (42%) | 25/43 (58%) | Reference | Reference | Reference | Reference | Reference | Reference | Reference | Reference |
| No | 80/294 (27%) | 214/294 (73%) | 3.02 (1.19–7.63) | 3.24 (1.15–9.18) | 3.12 (1.04–9.39) | 3.33 (1.12–9.94) | 3.14 (1.10–8.97) | 3.16 (1.05–9.53) | 3.42 (1.21–9.69) | 3.10 (1.09–8.80) |
| Unknown | 1/3 (33%) | 2/3 (67%) | 8.06 (0.17–386.04) | NA | NA | NA | NA | NA | NA | NA |
| **SARS-CoV-2 variant** | | | | | | | | | | |
| non-Alpha/Beta/Delta | 5/19 (26%) | 14/19 (74%) | 1.21 (0.15–9.96) | 1.19 (0.14–10.06) | 1.13 (0.12–10.46) | 1.27 (0.13–12.50) | 1.08 (0.12–9.43) | 1.17 (0.13–10.77) | 1.49 (0.18–12.64) | 1.08 (0.13–9.25) |
| Alpha | 2/17 (12%) | 15/17 (88%) | 3.19 (0.24–42.01) | 4.61 (0.31–69.26) | 3.79 (0.24–60.31) | 18.70 (0.95–368.10) | 3.19 (0.19–53.63) | 4.30 (0.28–65.69) | 5.42 (0.36–82.77) | 3.92 (0.26–60.08) |
| Beta | 70/230 (30%) | 160/230 (70%) | Reference | Reference | Reference | Reference | Reference | Reference | Reference | Reference |
| Delta | 8/50 (16%) | 42/50 (84%) | 3.00 (0.64–13.99) | 3.00 (0.60–14.89) | 2.78 (0.54–14.39) | 3.63 (0.72–18.44) | 2.29 (0.44–11.95) | 2.92 (0.57–14.94) | 3.40 (0.69–16.67) | 2.33 (0.45–12.09) |

*Table 6 continued on next page*

*Table 6 continued*

| | SARS-CoV-2 infection | | Univariate analysis | Multivariable analysis | Multivariable analysis (including contact parameter) | | | | | | |
|---|---|---|---|---|---|---|---|---|---|---|---|
| Variant Unknown | 14/24 (58%) | 10/24 (42%) | 0.48 (0.09–2.57) | 0.45 (0.08–2.48) | 0.45 (0.08–2.67) | 0.47 (0.08–2.72) | 0.37 (0.06–2.29) | 0.46 (0.08–2.56) | 0.46 (0.08–2.70) | 0.46 (0.09–2.44) | 0.46 (0.08–2.56) |
| Median daily duration* | 460 (165–1,250) | 680 (160–1,760) | 1.00 (1.00–1.00) | | 1.00 (1.00–1.00) | | | | | | |
| Maximum duration† | 39 (10–81) | 39 (13–96) | 0.99 (0.99–1.00) | | | 1.00 (0.99–1.00) | | | | | |
| Median average daily duration‡ | 33 (28–44) | 37 (28–47) | 0.98 (0.96–0.99) | | | | 0.98 (0.96–0.99) | | | | |
| Cumulative time in contact§ | 95 (46–198) | 127 (40–363) | 1.00 (1.00–1.00) | | | | | 1.00 (1.00–1.00) | | | |
| Median daily frequency¶ | 13 (5–27) | 18 (4–38) | 0.99 (0.99–1.00) | | | | | | 1.00 (0.99–1.01) | | |
| Maximum frequency** | 41 (15–83) | 46 (18–100) | 0.99 (0.99–1.00) | | | | | | | 1.00 (0.99–1.00) | |
| Daily average frequency†† | 2.4 (1.0–3.7) | 2.9 (1.0–6.8) | 1.01 (0.98–1.04) | | | | | | | | 1.03 (0.97–1.10) |

‡ n/ row N (%); Median (interquartile range).

*Median daily duration (median of cumulative duration of close-range proximity events for each day of deployment, in minutes).

†Maximum duration (longest duration of a close-range proximity event during deployment, in minutes).

‡Median average daily duration (median of cumulative duration of close-range proximity events in the day divided by the cumulative number of close-range proximity events during that day, in minutes).

§Cumulative time in contact (cumulative duration of close-range proximity events over the deployment period divided by the number of days sensor was worn, in minutes).

¶Median daily frequency (median of number of close proximity events for each day of deployment).

**Maximum frequency (highest number of close proximity events in one day during deployment).

††Daily average frequency (cumulative duration of close-range proximity events over the deployment period divided by the cumulative number of close-range proximity events during the deployment period). aOR: adjusted odds ratio. Significant associations on multivariable analysis in boldface. Factors investigated but not found significant on multivariable analysis: sex, HIV status, and underlying conditions.

## Acknowledgements

This research was funded by the Wellcome Trust (Grant number 221003/Z/20/Z) in collaboration with the Foreign, Commonwealth, and Development Office, United Kingdom. LD and CCa acknowledge support from the Lagrange Project of ISI Foundation funded by CRT Foundation, from the European Union's Horizon 2020 research and innovation program under grant agreement No. 101016233 (PERI-SCOPE). For the purpose of open access, the author has applied a CC BY-ND public copyright license to any Author Accepted Manuscript version arising from this submission.

## Additional information

### Group author details

**SA-S-HTS Group**
**Amelia Buys**: Centre for Respiratory Diseases and Meningitis, National Institute for Communicable Diseases of the National Health Laboratory Service, Johannesburg, South Africa; **Daniel G Amoako**: Centre for Respiratory Diseases and Meningitis, National Institute for Communicable Diseases of the National Health Laboratory Service, Johannesburg, South Africa; School of Health Sciences, College of Health Sciences, University of KwaZulu-Natal, KwaZulu-Natal, South Africa; **Dylan Toi**: Centre for Respiratory Diseases and Meningitis, National Institute for Communicable Diseases of the National Health Laboratory Service, Johannesburg, South Africa; **Jinal N Bhiman**: Centre for Respiratory Diseases and Meningitis, National Institute for Communicable Diseases of the National Health Laboratory Service, Johannesburg, South Africa; School of Pathology, Faculty of Health Sciences, University of the Witwatersrand, Johannesburg, South Africa; **Juanita Chewparsad**: Perinatal HIV Research Unit (PHRU), University of the Witwatersrand, Johannesburg, South Africa; **Kedibone Ndlangisa**: Centre for Respiratory Diseases and Meningitis, National Institute for Communicable Diseases of the National Health Laboratory Service, Johannesburg, South Africa; **Leisha Genade**: Perinatal HIV Research Unit (PHRU), University of the Witwatersrand, Johannesburg, South Africa; **Limakatso Lebina**: Perinatal HIV Research Unit (PHRU), University of the Witwatersrand, Johannesburg, South Africa; Africa Health Research Institute, Durban, South Africa; **Linda de Gouveia**: Centre for Respiratory Diseases and Meningitis, National Institute for Communicable Diseases of the National Health Laboratory Service, Johannesburg, South Africa; **Mzimasi Neti**: Centre for Respiratory Diseases and Meningitis, National Institute for Communicable Diseases of the National Health Laboratory Service, Johannesburg, South Africa; **Retshidisitswe Kotane**: Centre for Respiratory Diseases and Meningitis, National Institute for Communicable Diseases of the National Health Laboratory Service, Johannesburg, South Africa

### Competing interests

Jackie Kleynhans: reports institutional grant funding from UK Foreign, Commonwealth and Development Office, Wellcome Trust, WHO AFRO, Africa Pathogen Genomics Initiative (Africa PGI) through African Society of Laboratory Medicine (ASLM) and Africa CDC, US Centers for Disease Control and Prevention, South African Medical Research Council (SAMRC) and UK Department of Health and Social Care and managed by the Fleming Fund: SEQAFRICA project. The author has no other competing interests to declare. Lorenzo Dall'Amico: received support from the European Union's Horizon 2a020 research and innovation programme under grant agreements No. 101003688 (EpiPose) and No. 101016233 (PERISCOPE). The author acknowledges support from the Lagrange Project of ISI Foundation funded by CRT Foundation. They were awarded a PhD contract with Grenoble INP, and received support for attending meetings/travel from the ISP Foundation and Grenoble INP. The author has no other competing interests to declare. Laetitia Gauvin: has received Payment or honoraria for lectures from the University of Torino, Italy. They received support from the Lagrange Project of ISI Foundation funded by CRT Foundation. The authors has no other competing interests to declare. Michele Tizzoni: acknowledges support from the Lagrange Project of ISI Foundation funded by CRT Foundation. Sibongile Walaza: reports institutional grant funding from WHO AFRO, Africa Pathogen Genomics Initiative (Africa PGI) through African Society of Laboratory Medicine (ASLM) and Africa CDC, US Centers for Disease Control and Prevention, South African Medical Research Council (SAMRC), UK Department of Health and Social Care and managed by the Fleming Fund: SEQAFRICA project and the

Wellcome Trust. The author has no other competing interests to declare. Neil A Martinson: received grant funds from Pfizer. They participated on the data safety monitoring board of a TB HDT trial, and on the Scientific Advisory Board of a trial of an electronic reminder to take daily TB treatment. They hold an unpaid leadership role at the Setshaba Research Center. The authors has no other competing interests to declare. Anne von Gottberg: reports receiving grant funds from Sanofi Pasteur and Bill & Melinda Gates Foundation. They act as chairperson of the National Advisory Group on Immunisation. The author has no other competing interests to declare. Nicole Wolter: received grants from Sanofi Pasteur and Bill & Melinda Gates Foundation, and from US Centers for Disease Control and Prevention. The author has no other competing interests to declare. Cheryl Cohen: has received grant support from Sanofi Pasteur, US Centers for Disease Control and Prevention, Welcome Trust, Programme for Applied Technologies in Health (PATH), Bill & Melinda Gates Foundation and South African Medical Research Council (SA-MRC). The author has no other competing interests to declare. Ciro Cattuto: received support from the European Union's Horizon 2020 research and innovation programme under grant agreements No. 101003688 (EpiPose) and No. 101016233 (PERISCOPE), and from Fondation Botnar, EPFL COVID-19 Real Time Epidemiology I-DAIR Pathfinder. They also received support from the Lagrange Project of ISI Foundation funded by CRT Foundation, and received equipment from the European Space Agency. They were issued the patent US8660490B2. They received support for attending meetings/travel from Gulbenkian Foundation, and participate as a Steering Board Member at CRT Foundation, on the Advisory Board at The Data Guild and on the Advisory Board in GovLab's 100 Questions Initiative and Open Data Policy Lab. The authors has no other competing interests to declare. SA-S-HTS Group: The other authors declare that no competing interests exist.

### Funding

| Funder | Grant reference number | Author |
| --- | --- | --- |
| Wellcome Trust | 221003/Z/20/Z | Cheryl Cohen |
| Horizon 2020 Framework Programme | 101016233 | Ciro Cattuto |

The funders had no role in study design, data collection and interpretation, or the decision to submit the work for publication. For the purpose of Open Access, the authors have applied a CC BY public copyright license to any Author Accepted Manuscript version arising from this submission.

### Author contributions

Jackie Kleynhans, Conceptualization, Data curation, Formal analysis, Investigation, Methodology, Writing – original draft, Writing – review and editing; Lorenzo Dall'Amico, Laetitia Gauvin, Michele Tizzoni, Conceptualization, Formal analysis, Methodology, Writing – review and editing; Lucia Maloma, Data curation, Investigation, Methodology, Project administration, Writing – review and editing; Sibongile Walaza, Conceptualization, Resources, Formal analysis, Investigation, Methodology, Writing – review and editing; Neil A Martinson, Conceptualization, Resources, Investigation, Project administration, Writing – review and editing; Anne von Gottberg, Conceptualization, Resources, Investigation, Methodology, Writing – review and editing; Nicole Wolter, Conceptualization, Investigation, Methodology, Writing – review and editing; Mvuyo Makhasi, Conceptualization, Writing – review and editing; Cheryl Cohen, Conceptualization, Resources, Formal analysis, Supervision, Funding acquisition, Investigation, Methodology, Writing – review and editing; Ciro Cattuto, Conceptualization, Resources, Formal analysis, Supervision, Methodology, Writing – review and editing; Stefano Tempia, Conceptualization, Formal analysis, Supervision, Funding acquisition, Methodology, Writing – review and editing

### Author ORCIDs

Jackie Kleynhans  http://orcid.org/0000-0001-7081-6273
Lorenzo Dall'Amico  http://orcid.org/0000-0002-7493-6421
Anne von Gottberg  http://orcid.org/0000-0002-0243-7455

### Ethics

The study protocol was reviewed and approved by the University of the Witwatersrand Human Research Ethics Committee (Reference M2008114). The study was structured in accordance with the

Declaration of Helsinki. Written informed consent was obtained from all household members aged ≥18 years; assent was obtained from children aged 7-17 years, and consent from a parent/ guardian of children aged <18 years. Informed consent was administered by a study team member who explained the study and requirements to participants, and shared an information leaflet with participants to keep. Consent included the enrolment into the study and all required procedures, as well as the anonymous processing of personal data for the final study report. Participants in follow-up received grocery store vouchers of USD 3 per visit to compensate for the time required for specimen collection and interview, and an additional voucher once proximity sensors were returned with no visible damage.

### Decision letter and Author response
Decision letter https://doi.org/10.7554/eLife.84753.sa1
Author response https://doi.org/10.7554/eLife.84753.sa2

## Additional files

### Supplementary files
• MDAR checklist

### Data availability
The contact network, selected individual characteristics and analysis script are available at https://github.com/crdm-nicd/sashts (copy archived at *Kleynhans, 2023*).

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
