## [Editor Report]

This important study examines the association between SARS-CoV-2 infection and close contact among household members. The authors provide solid evidence that transmission of SARS-CoV-2 within households is not dependent upon close contact. The observations and analyses presented here raise important questions about the mechanics of respiratory pathogen transmission and should inspire future work.

---

## [Decision Letter]

**Decision letter after peer review:**

Thank you for submitting your article "Association of close-range contact patterns with SARS-CoV-2: a household transmission study" for consideration by *eLife*. Your article has been reviewed by 3 peer reviewers, one of whom is a member of our Board of Reviewing Editors, and the evaluation has been overseen by Neil Ferguson as the Senior Editor. The following individual involved in the review of your submission has agreed to reveal their identity: Virginia E. Pitzer (Reviewer #3).

Essential revisions:

This is a strong and generally well-presented study. We request revisions to help strengthen the impact of the work.

1) Please see the numerous small questions and points of confusion described by all three reviewers. Please revise the manuscript accordingly, unless you believe the peer review (which included cross-review) reflects an unusual degree of misreading.

2) Please consider some of the alternative metrics for close contacts suggested by reviewers 2 and 3 or expand the discussion of how alternate contact parameters might lead to different results, potentially supporting a greater role for droplet transmission (or not).

3) Please choose more reasonable units for the ORs (see reviewer 3) and interpret them in the text.

*Reviewer #1 (Recommendations for the authors):*

Excellent job making the code and data available.

It was not clear to me when seropositive individuals were excluded if they truly "disappeared" from the analysis, e.g., excluded from the household size statistics.

As I alluded to in the public review, I'm not clear on what adjustments for age (or even variant) are really doing here. Gender presumably could be a proxy of time at home. How much confidence is there in the identification of previously infected individuals? As I also alluded to in the public review, I am curious if the impact of proximity becomes more pronounced in wholly susceptible households, or if any of the other associations change substantially (although I recognize there will be less power).

I was surprised to read the statement that droplet transmission is "thought to be the most important [respiratory transmission] route." Maybe (infamously) by the WHO? The claim seems to ignore major work by Jimenez, Prather, Marr, Milton et al. by the time of submission; early outbreak investigations (planes, workplaces, etc.); and decades of prior work demonstrating a very large role for aerosol transmission with measles, tuberculosis, influenza, etc. I do not think the study would at all be weakened by acknowledging the strong evidence for aerosol transmission.

Ll. 153-154: I was confused by the 5-min time slice exclusion. If people were still, the contact was excluded.

l. 195: I found "(number of nodes)" confusing.

*Reviewer #2 (Recommendations for the authors):*

I have the following comments and suggestions:

1. On the 3rd page, Methods, the authors mentioned "The resulting total sample size was 440 exposed household members". However, on the 6th page, Results, the authors mentioned "identified 277 (18%) positive for SARS-CoV-2, of which 124 (45%) were enrolled and included in the household cumulative infection risk analysis".

This comparison may suggest that the actual number of study participants was much smaller than the target sample size. The authors may need to clarify this.

2. Lines 167 – 177, the authors mainly considered three types of contact parameters: including (1) duration (median daily cumulative time in contact in seconds), (2) frequency (median daily number of contacts with the index/infected individuals over the deployment period), and (3) average duration (cumulative time in contact divided by the cumulative number of close-range proximity events over full deployment period).

I'm not sure if the median measurements of the contact duration, frequency, and average duration are representative of the daily human behaviour within the household settings. For example, the duration of a formal lunch or dinner can be longer than one hour. Could you include more contact parameters such as long-duration contacts in your analysis?

3. It's also not clear if the household members could contact other people within the same community. For example, did you require the study participants to stay at home during the study's 2-week period? Could you adjust the community force of infection in your regression model?

4. In lines 109-110, the authors mentioned "with {greater than or equal to}2 household contacts of whom none reported symptoms prior to index case onset". It's not clear why the authors needed this requirement.

5. In lines 163-165, the authors mentioned "For the analysis, we only considered close-range proximity events that occurred one day after deployment and one day before collection".

Do you mean the proximity events occurred between deployment and collection?

---

## [Author Response]

Essential revisions:This is a strong and generally well-presented study. We request revisions to help strengthen the impact of the work.

We thank the editors for the positive feedback.

1) Please see the numerous small questions and points of confusion described by all three reviewers. Please revise the manuscript accordingly, unless you believe the peer review (which included cross-review) reflects an unusual degree of misreading.

Please see our responses to individual comments below in green. Line numbers refer to the tracked version of the manuscript.

2) Please consider some of the alternative metrics for close contacts suggested by reviewers 2 and 3 or expand the discussion of how alternate contact parameters might lead to different results, potentially supporting a greater role for droplet transmission (or not).

We added four additional parameters: The maximum frequency and duration of close contact events – to capture the extreme end of the spectrum; the daily average close proximity event frequency (cumulative number of events / deployment days), and the cumulative time in contact (normalised by deployment days).

3) Please choose more reasonable units for the ORs (see reviewer 3) and interpret them in the text.

All time-based close proximity event contact parameters are now displayed in minutes, instead of seconds.

Reviewer #1 (Recommendations for the authors):Excellent job making the code and data available.It was not clear to me when seropositive individuals were excluded if they truly "disappeared" from the analysis, e.g., excluded from the household size statistics.

To clarify, we updated the final sentence of the Screening, enrolment and follow-up Methods section (line 130-132, page 4): “Individuals were considered seropositive if they tested positive on either assay. Individuals sero-positive at the start of follow-up with no rRT-PCR confirmed SARS-CoV-2 infection during follow-up were excluded from the risk factor analysis for household SARS-CoV-2 acquisition as they may have been protected from infection, but were still considered in the household size parameter.”

As I alluded to in the public review, I'm not clear on what adjustments for age (or even variant) are really doing here. Gender presumably could be a proxy of time at home. How much confidence is there in the identification of previously infected individuals? As I also alluded to in the public review, I am curious if the impact of proximity becomes more pronounced in wholly susceptible households, or if any of the other associations change substantially (although I recognize there will be less power).

We added another sensitivity analysis where any households where individuals were excluded due to seropositivity was excluded from the analysis. The results (Table 5) showed that none of the contact patterns were significantly associated with SARS-CoV-2 transmission in the household.

I was surprised to read the statement that droplet transmission is "thought to be the most important [respiratory transmission] route." Maybe (infamously) by the WHO? The claim seems to ignore major work by Jimenez, Prather, Marr, Milton et al. by the time of submission; early outbreak investigations (planes, workplaces, etc.); and decades of prior work demonstrating a very large role for aerosol transmission with measles, tuberculosis, influenza, etc. I do not think the study would at all be weakened by acknowledging the strong evidence for aerosol transmission.

Updated statement as follows (Introduction, line 70-71, page 3): “SARS-CoV-2 transmission is mainly via the respiratory route, through both droplet-mediated and airborne transmission.”

Ll. 153-154: I was confused by the 5-min time slice exclusion. If people were still, the contact was excluded.

Added line 157-157, page 5 to clarify: “Accelerometers are very sensitive and even a slight movement will be detected, therefore contacts that occurred while individuals were sitting/standing still will still be included.”

l. 195: I found "(number of nodes)" confusing.

Fragment of sentence removed, line 212 page 6: “For the analysis with close-range proximity events with all SARS-CoV-2 infected household members, we included an offset term in the model to account for the number of SARS-CoV-2 infected members in contact with.

Reviewer #2 (Recommendations for the authors):I have the following comments and suggestions:1. On the 3rd page, Methods, the authors mentioned "The resulting total sample size was 440 exposed household members". However, on the 6th page, Results, the authors mentioned "identified 277 (18%) positive for SARS-CoV-2, of which 124 (45%) were enrolled and included in the household cumulative infection risk analysis".This comparison may suggest that the actual number of study participants was much smaller than the target sample size. The authors may need to clarify this.

We have acknowledged this in the limitations: “Discussion, line 345-347, page 9: “The small sample size may have reduced our power to detect small differences in close proximity event parameters between those infected with SARS-CoV-2 and those not infected.”

2. Lines 167 – 177, the authors mainly considered three types of contact parameters: including (1) duration (median daily cumulative time in contact in seconds), (2) frequency (median daily number of contacts with the index/infected individuals over the deployment period), and (3) average duration (cumulative time in contact divided by the cumulative number of close-range proximity events over full deployment period).I'm not sure if the median measurements of the contact duration, frequency, and average duration are representative of the daily human behaviour within the household settings. For example, the duration of a formal lunch or dinner can be longer than one hour. Could you include more contact parameters such as long-duration contacts in your analysis?

We added four additional parameters: The maximum frequency and duration of close-range proximity events – to capture the extreme end of the spectrum; the daily average close-range proximity event frequency (cumulative number of events / deployment days), and the cumulative time in contact (normalised by deployment days). The methods section, line 173-185 (page 5) has been updated with the details: “We assessed the following close-range proximity event parameters: (1) median daily duration (median of cumulative duration of close-range proximity events for each day of deployment, in minutes), (2) maximum duration (longest duration of a close-range proximity event during deployment, in minutes), (3) median frequency (median of number of close proximity events for each day of deployment), (4) maximum frequency (highest number of close proximity events in one day during deployment), (5) median average daily duration (median of cumulative duration of close-range proximity events in the day divided by the cumulative number of close-range proximity events during that day), (6) daily average contact frequency (cumulative duration of close-range proximity events over deployment period divided by the cumulative number of close-range proximity events during the deployment period), and (7) cumulative time in contact (cumulative duration of close-range proximity events over deployment period divided by number of days sensor worn).”

3. It's also not clear if the household members could contact other people within the same community. For example, did you require the study participants to stay at home during the study's 2-week period? Could you adjust the community force of infection in your regression model?

To clarify on participants staying at home during the period we added to the Discussion section line 331-334, page 8-9: “Furthermore, although based on legislation, close contacts (including household contacts) of SARS-CoV-2 cases were supposed to quarantine, compliance was not monitored. Therefore, household contacts could have been exposed to non-household SARS-CoV-2 cases during the follow-up period.”

Our current analytic framework does not allow for the adjustment of the community force of infection. We will be considering this in future analyses.

4. In lines 109-110, the authors mentioned "with {greater than or equal to}2 household contacts of whom none reported symptoms prior to index case onset". It's not clear why the authors needed this requirement.

We updated lines 108-112 (page 3) of the methods to clarify: “We enrolled household contacts of SARS-CoV-2 infected individuals identified through screening (presumptive index) with ≥2 household contacts (for efficient investigation of risk factors for transmission in the household, weighting cost of household visits and data collected) of whom none reported symptoms prior to index case onset (reducing probability of previous recent SARS-CoV-2 infection in the household).”

5. In lines 163-165, the authors mentioned "For the analysis, we only considered close-range proximity events that occurred one day after deployment and one day before collection".Do you mean the proximity events occurred between deployment and collection?

To clarify we updated line 168-171 (page 5) of the methods section: “For the analysis, we only considered close-range proximity events that occurred one day after sensors were issued and one day before collection, hence excluding any false events logged when sensors were prepared, handed out and collected in the household.”